# Future sea-level projections with a coupled atmosphere-ocean-ice-sheet model

Jun-Young Park [1,2] ✉, Fabian Schloesser [3] ✉, Axel Timmermann [1,4],
Dipayan Choudhury[1,5], June-Yi Lee [1,2,6] & Arjun Babu Nellikkattil[1,2]

Climate-forced, offline ice-sheet model simulations have been used extensively in assessing how much ice-sheets can contribute to future global sea-level rise. Typically, these model projections do not account for the two-way interactions between ice-sheets and climate. To quantify the impact of ice-ocean-atmosphere feedbacks, here we conduct greenhouse warming simulations with a coupled global climate-ice-sheet model of intermediate complexity. Following the Shared Socioeconomic Pathway (SSP) 1-1.9, 2-4.5, 5-8.5 emission scenarios, the model simulations ice-sheet contributions to global sea-level rise by 2150 of $0.2 \pm 0.01$, $0.5 \pm 0.01$ and $1.4 \pm 0.1$ m, respectively. Antarctic ocean-ice-sheet-ice-shelf interactions enhance future subsurface basal melting, while freshwater-induced atmospheric cooling reduces surface melting and iceberg calving. The combined effect is likely to decelerate global sea-level rise contributions from Antarctica relative to the uncoupled climate-forced ice-sheet model configuration. Our results demonstrate that estimates of future sea-level rise fundamentally depend on the complex interactions between ice-sheets, icebergs, ocean and the atmosphere.

Global mean sea-level (SL) has risen over the past century by about 20 cm, in part due to the thermal expansion of seawater, glacier and ice-sheet melt and changes in groundwater storage[1–3]. This trend is likely to accelerate in response to increasing atmospheric greenhouse gas concentrations and anthropogenic warming. With a considerable fraction of the world's population living near coastlines, it is crucial to provide accurate projections of global and regional future SL trends and constrain remaining uncertainties. The largest uncertainty originates from the response of the Antarctic ice-sheet (AIS) to greenhouse warming. Recent model-based estimates of the 21st century AIS contribution to future SL rise for a Representative Concentration Pathway (RCP) 8.5 scenario range from 0 to 1.4 m[3–8].

These estimates were obtained with offline ice-sheet models that use atmospheric and oceanic forcings from future climate model projections. Accordingly, several issues need to be considered: (i) the transient and equilibrium climate sensitivities of the current

generation of earth system models still have remaining uncertainties of 1.5–2.2 K and 2.6–4.1 K, respectively[9], (ii) several processes such as iceberg calving or basal melting are not well constrained and represented; also different ice-sheet models show varying sensitivities to warming scenarios[5,10,11], and (iii) the impact of climate-ice-sheet coupling is not included in offline ice-sheet model simulations, even though it may play an important role in Southern Hemisphere climate change[12,13]. Here, we present a new suite of coupled future earth system model simulations, which captures important interactions between atmosphere, ocean, ice-sheets, ice-shelves, and icebergs in both hemispheres.

Observational data shows that AIS meltwater discharge has increased over the past decades[14,15]. This in turn can increase Southern Ocean (SO) stratification and subsurface warming due to reduced vertical heat exchange. Subsurface Southern Ocean (SSO) warming enhances sub-shelf melting[16–19] which can lead to a reduction of the

[1]Center for Climate Physics, Institute for Basic Science, Busan, Republic of Korea. [2]Department of Climate System, Pusan National University, Busan, Republic of Korea. [3]International Pacific Research Center, University of Hawaii, Honolulu, Hawaii, USA. [4]Pusan National University, Busan, Republic of Korea. [5]Climate Change Research Centre, University of New South Wales, Sydney, NSW, Australia. [6]Research Center for Climate Sciences, Pusan National University, Busan, Republic of Korea. ✉e-mail: junyoung1989@pusan.ac.kr; schloess@hawaii.edu

buttressing effect of ice-shelf on grounded ice[20,21]. As a consequence the flow of ice streams can accelerate toward the ocean[22], which would translate to SL rise. These processes can be further amplified by other feedbacks, including the Marine Ice-Sheet Instability (MISI) along retrograde slopes[23], hydrofracturing and the Marine Ice-Cliff Instability (MICI)[5]—all of which remain poorly constrained[8,10].

To quantify the effect of these interactions on future SL projections, one needs to employ coupled global climate-ice-sheet models[24,25], which capture the complex interactions between climate components and ice-sheet, ice-shelf and iceberg dynamics and thermodynamics. Our study focuses on the AIS and Greenland ice-sheet (GrIS) contributions to future SL projections using an ensemble of simulations conducted with the coupled three-dimensional climate-ice-sheet-iceberg modeling system[26] LOVECLIP (Supplementary Fig. S1). LOVECLIP is based on the climate model of intermediate complexity LOVECLIM[27] and the Penn State University Ice-Sheet model (PSUIM)[5,28,29]. Biases of surface air temperature, precipitation, and SSO are corrected from LOVECLIM to PSUIM[26]. PSUIM is forced by surface air temperature, precipitation, evaporation, solar radiation, and annual mean subsurface ocean temperature. Surface air temperature and precipitation are downscaled vertically to PSUIM grid with applied lapse rate corrections[28].

The simulations include an 8000-years-long coupled pre-industrial spin-up run for initialization and 10 member ensemble of simulations forced by increasing $CO_2$ concentrations following the Shared Socioeconomic Pathway (SSP) 1–1.9, 2–4.5 and 5–8.5 scenarios[30] until 2150 CE. To further elucidate the effect of AIS meltwater flux on polar climate, the stability of the ice-sheets and SL we performed an additional idealized sensitivity experiment (Supplementary Table S1) for which we scaled the amplitudes of AIS liquid runoff and iceberg calving to balance net precipitation over Antarctica (experiment SSP5-8.5_MWOFF). In addition, to quantify the impact of AIS hydrofracturing and ice-cliff failure on the ice-sheet evolution, these parameterizations are turned off (CREVLIQ = 0 m per (m/year)$^{-2}$ and VCLIF = 0 km/year) with and without meltwater flux coupling (experiments SSP5-8.5_HFCMOFF and SSP5-8.5_MWHFCMOFF). Furthermore, to estimate the sensitivity of Antarctic ice-shelf mass loss to SSO warming, we doubled the sub-shelf SSO temperature anomaly (relative to 1850 CE) in the SSP5-8.5 scenario with (Re_SSP5-8.5_2xSOTA) and without meltwater fluxes (Re_SSP5-8.5_2xSOTA_MWOFF).

## Results
### Recent trend and interannual variability of GrIS and AIS
According to satellite observations, the GrIS and AIS have been losing mass at a rate of ~286 Gt/year in 2010–2018 CE and ~252 Gt/year in

2009–2017 CE, respectively[31,32]. Quantifying the natural and anthropogenic contributions to this trend remains difficult because of our limited understanding of naturally occurring low-frequency ice-sheet dynamics and the relatively short observational period. Here we compare the observed 19-year trend of ice mass balance from the Gravity Recovery and Climate Experiment (GRACE)[33] for the period 2002–2020 CE to the corresponding values in forced experiments as well as a 5000-year-long pre-industrial control run (CTR) (Supplementary Fig. S2) conducted with LOVECLIP (Fig. 1). In Fig. 1, each 19-year chunk of mass balance in CTR is cut after high-pass filtering over than 80 years and then, 19-year trends of nature variability are extracted. Those trends are expressed here in terms of sea-level-equivalent (SLE, 1 m SLE = 3.62 × 10$^{14}$ m$^3$). Consistent with the GRACE measurements, changes in the mass balance are calculated only from the grounded parts of the ice-sheets for LOVECLIP. Interannual variability of the mass balance recorded by GRACE and simulated by the forced LOVECLIP experiments during 2002–2020 CE fall within the range of natural variability exhibited by the CTR (Supplementary Fig. S3). This indicates that the range of interannual mass balance changes of ice-sheets are represented realistically in the model simulation on the global scale. However, the observed GrIS 19-year trend (−0.075 cm/year SLE) lies outside the respective 95% confidence interval range of CTR (Fig. 1a), which suggests that the current observed mass loss in Greenland is inconsistent with natural variability, as estimated from LOVECLIP. Although the simulated GrIS trend (−0.13 to −0.08 cm/year SLE) is slightly overestimated than the observed GRACE estimate for the same period (Fig. 1a, red range), we can still conclude that greenhouse warming contributed to GrIS melting over the past decades. On the other hand, the AIS mass balance trend recorded by GRACE (−0.04 cm/year SLE) and the forced AIS trend (−0.1 to −0.02 cm/year SLE) lie within the 95% confidence range of the LOVECLIP CTR simulation due to the fact that AIS natural variability amplitude exceeds that of the GrIS by a factor of 7.

### Future change of global surface temperature and SL
The projected ensemble average of global surface temperature rise in 2100 CE (2150 CE) relative to the pre-industrial levels (1850–1900 CE) amounts to 1.4 ± 0.17 °C (1.2 ± 0.14 °C), 2.4 ± 0.15 °C (2.7 ± 0.16 °C) and 4.0 ± 0.15 °C (5.3 ± 0.09 °C) for the SSP1-1.9, SSP2-4.5, and SSP5-8.5 scenarios, respectively (Fig. 2a). The uncertainty values are calculated at 95% confidence interval in this paper. Relative to the recent past (1995–2014 CE) the simulated end-of-century warming (2081–2100 CE) attains values of 0.3 °C for SSP1-1.9 and 2.6 °C for SSP5-8.5, which is at the lower end of the multi-model range in projected

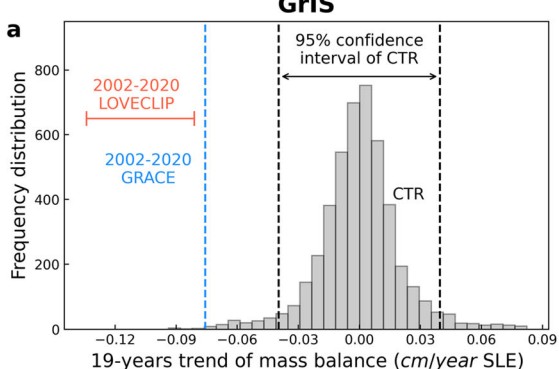
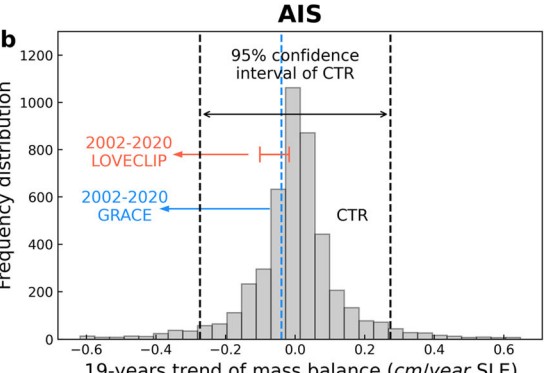

**Fig. 1 | 19-year trends of observed and simulated mass balance of Greenland ice-sheet (GrIS) and Antarctic ice-sheet (AIS). a** Histogram of each extracted 19-year trend of Greenland mass balance after 80-year-high-pass filtering in the 5000-year-long pre-industrial control run (CTR, gray histogram) with 95% confidence interval range of CTR (black dashed line), and observed estimates of 19-year trend for 2002–2020 CE from the Gravity Recovery and Climate Experiment (GRACE)[33] (blue dashed line) and simulated by the forced LOVECLIP ensemble (red line) in sea-level-equivalent (SLE); **b** same as **a**, but for Antarctica. Consistent with the GRACE measurements, mass balance terms for LOVECLIM are calculated in this figure using only the grounded ice-sheet portion.

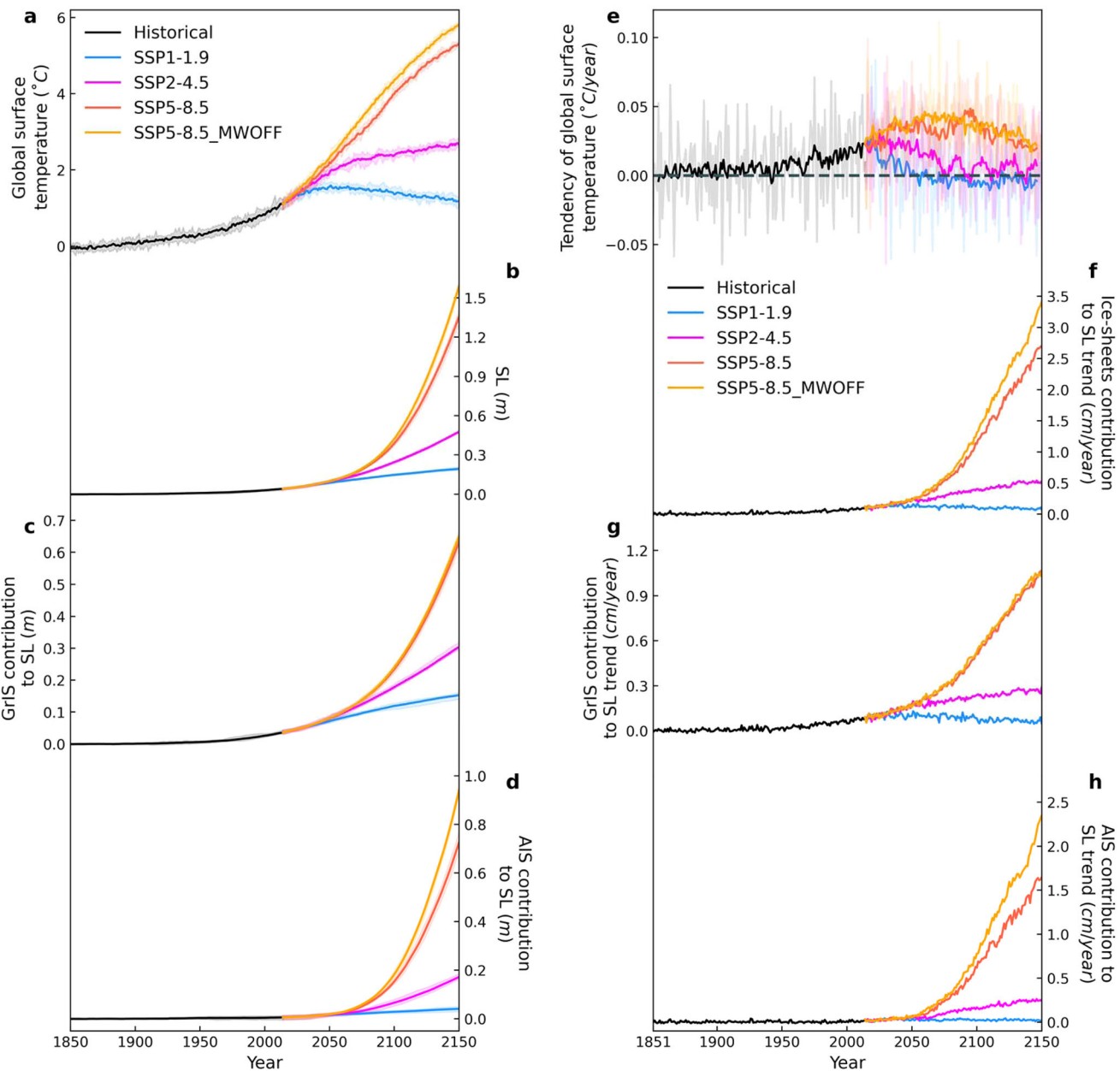

**Fig. 2 | Global surface temperature and sea-level (SL) projections, and their tendencies. a–d** Annual anomalies (relative to the 1850–1900 CE mean) of (**a**) the global surface temperature, (**b**) SL, and (**c**) SL contributions from the Greenland ice-sheet (GrIS) and (**d**) Antarctic ice-sheet (AIS). **e–h** are the respective time derivatives of **a–d** (change per year). Solid lines of **a–d** indicate the ensemble mean and

shading the ensemble range. The solid line in **e** represents the 9-year moving average of the time derivative of global surface temperature, with the dashed line indicating 0 °C/year. Different colors represent the historical (black line; period 1850–2014 CE), and SSP1-1.9 (blue line), SSP2-4.5 (pink line), SSP5-8.5 (red line) and SSP5-8.5_MWOFF (orange line) simulations during the period 2014–2150 CE.

changes obtained from the respective Coupled Model Inter-comparison 6 (CMIP6) models[34,35].

Higher surface temperatures increase ice-sheet surface melting and subsequent meltwater discharge, and ice-sheet calving in both hemispheres. For the SSP1-1.9, SSP2-4.5 and SSP5-8.5 scenarios the GrIS contributes about 12 ± 1, 18 ± 0.9 and 23 ± 1.6 cm and the AIS adds 3 ± 0.8, 7 ± 1.4, and 15 ± 1.5 cm to SL by the year 2100 relative to pre-industrial levels (Fig. 2c, d). 2100 CE (2150 CE) LOVECLIP simulates for the respective scenarios a total ice-sheet contribution to SL of 15 ± 0.9, 24 ± 1.3, 39 ± 2 (19 ± 1.4, 48 ± 1.4, 136 ± 6.2) cm (Fig. 2b). The GrIS and AIS contributions lie within the range of estimates obtained from uncoupled scenario-forced models for Greenland[36–38] and Antarctica[6,8,39,40]. One factor impacting the LOVECLIP ice-sheet response is the relatively weak temperature sensitivity to greenhouse

forcing compared to most CMIP6 models (Supplementary Fig. S8). With lower sensitivity, nonetheless, our LOVECLIP shows both Arctic and Antarctic amplification. On the other hand, CMIP6 models do not show the aspect of Antarctic amplification. From a climate sensitivity point of view, our model results can therefore be regarded as con-servative estimates. Our simulated SL is also substantially lower than the projected 1 m end-of-century AIS contribution to SL presented in a series of offline ice-sheet model simulations conducted with the PSUIM[5]. In our coupled model simulations, which use the same ice-sheet model, but a different climate model, different ice-sheet and coupling parameters and at lower resolution, the rate of global tem-perature change slows down (negative second derivative with respect to time) around 2100 CE for SSP2-4.5 and SSP5-8.5 (Fig. 2a). This strongly contrasts the continued acceleration (positive second

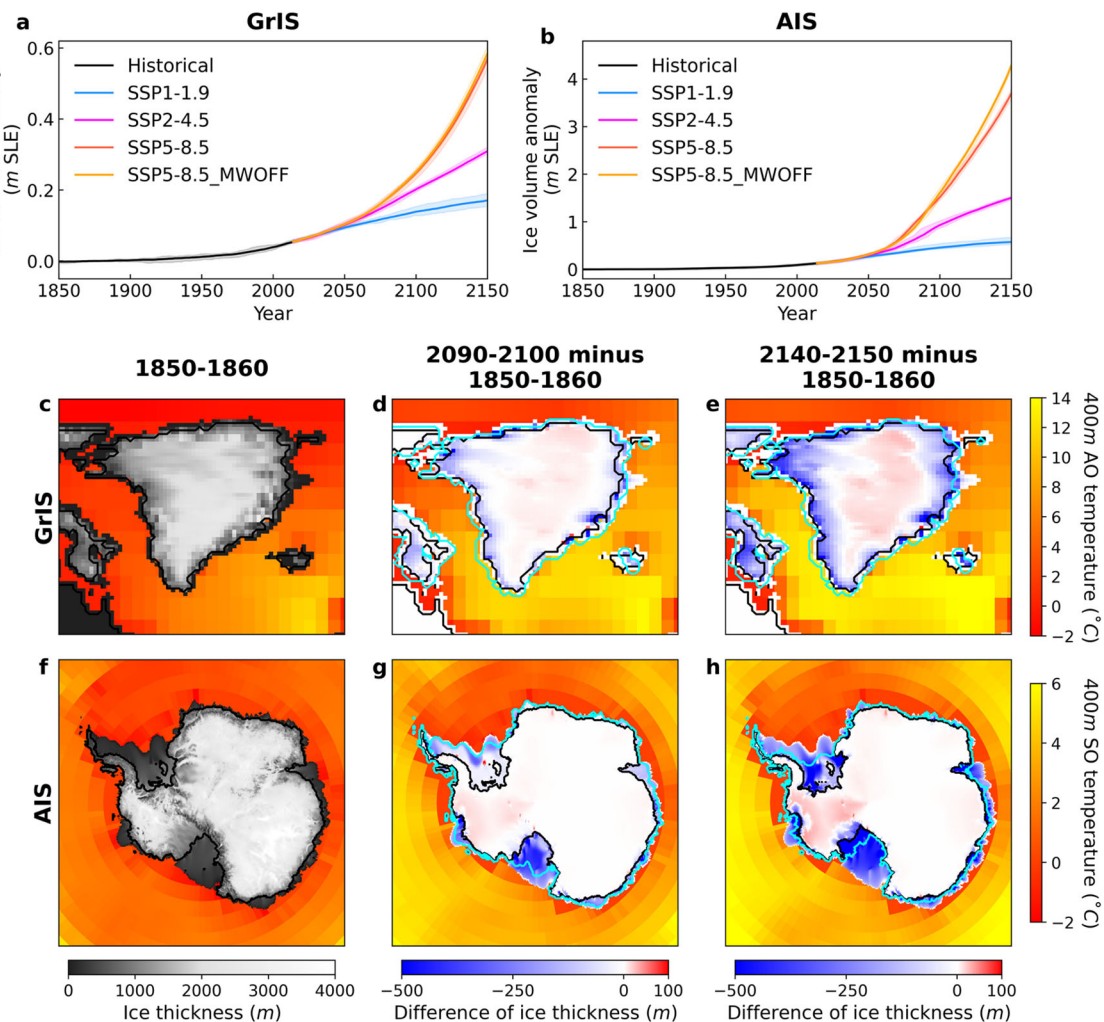

**Fig. 3 | Projected changes in mass balance, ice thickness and subsurface ocean temperature. a**, **b** Time series of the annual mean (**a**) Greenland ice-sheet (GrIS) and (**b**) Antarctic ice-sheet (AIS) net mass balance in sea-level-equivalent (SLE) (including contributions from ice shelves), respectively; **c** GrIS 1850–1860 CE mean ice thickness (grayscale colormap) and 400 m Arctic Ocean (AO) temperature (red-yellow colormap); **d** 2090–2100 CE change in SSP5-8.5 scenario of GrIS ice thickness with respect to 1850–1860 CE and change in mean AO subsurface temperature; **e** same as **d**, but for period 2140–2150 CE; **f**, **g** same as **c**–**e**, but for AIS with 400 m Southern Ocean (SO) temperature. Black contours indicate simulated grounding lines for different periods. Cyan contours indicate the edge lines of ice-shelves for different periods.

derivative) of SL (Fig. 2b) for these scenarios. This behavior illustrates the combined effect of long response timescales of the ice-sheets, the effect of positive feedbacks and their prolonged contribution to SL, even long after $CO_2$ emissions have started to decline. Reductions in future greenhouse gas emissions can help slowdown global warming trends for the high-end emission scenarios (Fig. 2e). However, they are unlikely to stop the ice-sheet-driven SL rise acceleration (Fig. 2f–h) and the apparent run-away in SL for the next 130 years. Only the much more aggressive SSP1-1.9 scenario can lead to a gradual slow-down of SL rise acceleration (Fig. 2f), which implies that, according to our simulations, the 2°C warming (above the pre-industrial level) target emphasized by the Paris agreement[41] is insufficient to prevent accelerated SL rise over the next century[42].

## Ice loss from GrIS and AIS

In our model simulations, future warming leads to an increase in snow accumulation and ice thickness in the central part of GrIS (Fig. 3c–e, Fig. 4a) and West Antarctica (Fig. 3f–h, Fig. 4e). However, the negative mass balance terms together are considerably larger (Fig. 4b–d, f–h), leading to a net projected 80-year mass loss for the different scenarios of 14 ± 1.5, 20 ± 0.9 and 25 ± 1.5 cm SLE for GrIS and 46 ± 6, 94 ± 8 and 152 ± 8 cm for AIS, respectively (Fig. 3a, b). Although the AIS shows significantly more ice melting until 2100 CE in comparison to the GrIS, its contribution to SL is similar or even lower (Fig. 2c, d) because most of the GrIS melting occurs at the surface as ice ablation (Fig. 3d, e, Fig. 4b), whereas the AIS loses mass primarily below-SL and at the shelves through melting and calving, especially in the Ross ice-shelf and Ronne-Filchner ice-shelf regions (Fig. 3g, h, Fig. 4g, h). Ice-shelf and submarine ice-loss do not directly contribute to SL rise (except for marginal contributions from the difference in ice and seawater density). In our simulations warmer Circumpolar Deep Water (CDW) reaches the continental shelf regions which in turn increases basal melting, sub-shelf melting and potential grounding line retreat (Fig. 3g, h). Due to the larger extent of ice-shelves, basal melting is more important for the AIS than for the GrIS (Fig. 4c, g). Note that global coarse-resolution ocean models, such as the one used here with a 3° × 3° degree horizontal resolution cannot fully resolve the small-scale coastal ocean circulation processes around Antarctica[43] and ignore sub-cavity flows, which are important to explicitly resolve basal melting processes. In our modeling framework basal melting is parameterized using open ocean temperatures interpolated on the finer ice-sheet model grid[28].

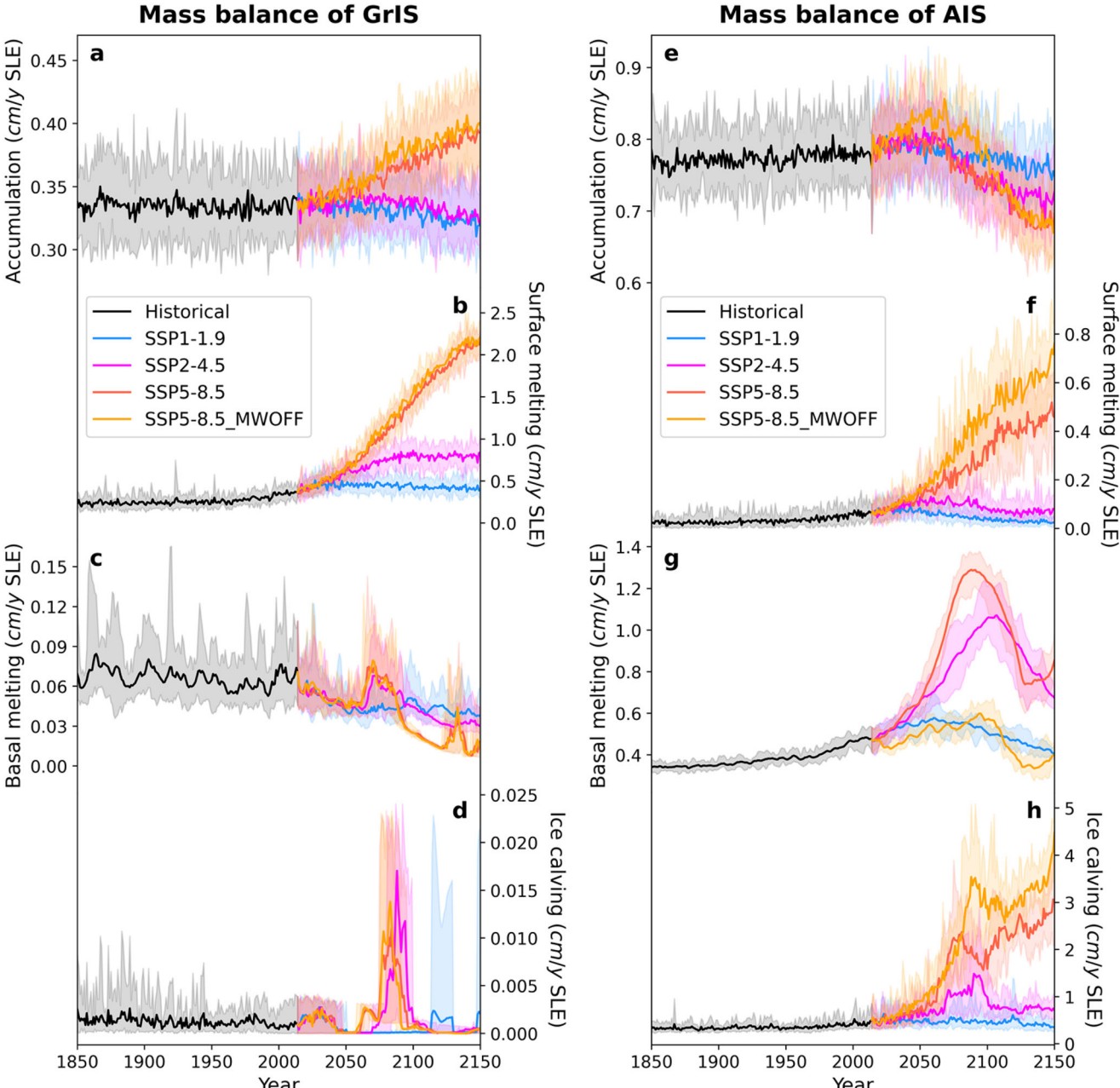

**Fig. 4 | Individual mass balance terms for Greenland ice-sheet (GrIS) and Antarctic ice-sheet (AIS). a–d** Represent the individual GrIS mass balance terms for (**a**) the accumulation, (**b**) surface melting, (**c**) basal melting and (**d**) ice calving expressed as sea-level-equivalent (SLE) per year; **e–h** same as **a–d**, but for AIS. Solid lines indicate the ensemble mean and shading the ensemble range. Different colors represent the historical (black line; period 1850–2014 CE), and SSP1-1.9 (blue line), SSP2-4.5 (pink line), SSP5-8.5 (red line) and SSP5-8.5_MWOFF (orange line) simulations during the period 2014–2150 CE.

According to our numerical experiments, the Ross ice-shelf completely disappears in the SSP5-8.5 scenario after 2100 CE (Fig. 3g, h). At this time basal melting and calving rates peak (Fig. 4g, h). A secondary simulated increase in these fluxes at the beginning of the 22$^{nd}$ century is associated with an accelerated retreat of the Ronne-Filchner ice-shelves (Figs. 3h, 4g, h). Even though the AIS contribution to SL rise is initially smaller than that of the GrIS (before 2100 CE), the rapid loss of stabilizing ice-shelves leads to a gradual increase of ice flow across the grounding lines that will initiate positive ice-sheet feedbacks associated with the MISI[23], hydrofracturing and MICI[5,29]. The AIS calving fluxes, which attain values of ~2 cm/year SLE by 2080 CE (corresponding to a freshwater flux into the ocean of ~0.34 Sv; 1 Sv = $10^6$ m$^3$/s), dominate the negative mass balance and global SL contribution. The accelerated

mass loss over the AIS is related to a combination of surface melting, basal melting and grounding line retreat which contributes to the massive ice calving fluxes (Fig. 4f–h) – each component with their individual temporal contributions to the total freshwater and SL effect.

In contrast to the AIS, the GrIS shows a gradual decrease in basal melting and ice calving fluxes (Fig. 4c, d), interrupted only by an abrupt GrIS ice calving event around 2090 CE in SSP5-8.5, which is associated with a complete loss of small ice-shelf areas. In Greenland the dominant mass loss and the contribution to SL are due to the positive trend in surface melting, which attains values of up to 2.1 ± 0.3 cm/year SLE by 2150 CE (Fig. 4b)—a ~30-fold increase compared to the recent observed interannual rates of GrIS mass loss (Supplementary Fig. S3a).

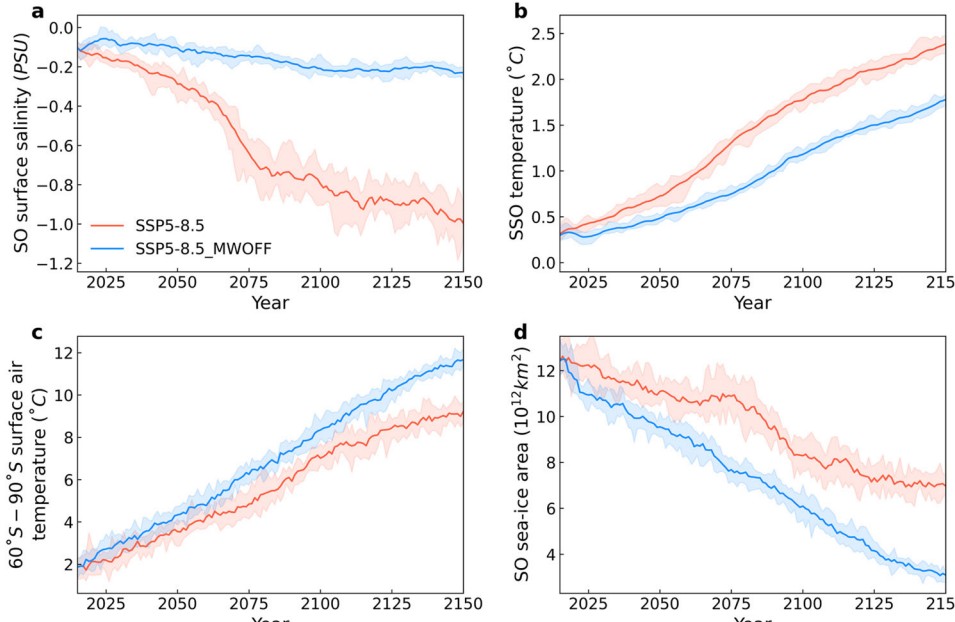

**Fig. 5 | Climate-ice-sheet feedbacks in Southern Hemisphere. a–c** Annual anomalies (relative to the 1850–1900 CE mean) of (**a**) the Southern Ocean (SO) surface salinity, (**b**) 400 m subsurface Southern Ocean (SSO) temperature and (**c**) surface air temperature averaged between 60°S and 90°S. **d** is the SO sea-ice area averaged between 60°S and 90°S. Solid lines indicate the ensemble mean and shading the ensemble range. Different colors represent the SSP5-8.5 (red line) and SSP5-8.5_MWOFF (blue line) simulations during the period 2014–2150 CE.

## Ice-sheet/climate feedbacks in Southern Hemisphere

To further quantify the effects of climate-ice-sheet coupling in the Southern Hemisphere, and test the previously hypothesized positive CDW/MISI feedback[16–19,44] we performed idealized SSP5-8.5 ensemble sensitivity experiments in which the freshwater coupling from the Antarctic meltwater is decoupled (experiment SSP5-8.5_MWOFF). Increased AIS meltwater fluxes in the fully coupled model experiment (experiment SSP5-8.5) reduce surface ocean salinity in the SO relative to SSP5-8.5_MWOFF (Fig. 5a). In turn this increases ocean stratification and reduces vertical heat exchange between cold surface and warmer subsurface waters. As a result, annual mean subsurface temperatures around Antarctica increase by 1.5 °C over the 21st century in SSP5-8.5 (Fig. 5b). In contrast, in SSP5-8.5_MWOFF the AIS melting does not directly impact the SO stratification, which leads to a temporary 30% reduction in subsurface ocean warming (Fig. 5b) and a 50% reduction in basal melting (Fig. 4g). At the surface, increased stratification in SSP5-8.5 and reduced vertical heat exchange lead to cooling and increased sea-ice production[6,12,13,45] (Fig. 5d), relative to the SSP5-8.5_MWOFF experiment. The 21st century annual mean surface air temperatures around Antarctica are about 1.4 °C colder in SSP5-8.5 as compared to SSP5-8.5_MWOFF (Fig. 5c). This cooling effect provides a negative feedback for AIS surface melting[6,42]. Moreover, without meltwater coupling temperatures, precipitation and snow accumulation increase over Antarctica by about 0.1 cm/year SLE (Fig. 4e) around 2100 CE. At the ice-sheet margins, higher temperatures and increased precipitation in SSP5-8.5_MWOFF contribute to hydrofracturing and the simulated increased calving rates[5] (Fig. 4h). Overall, in the fully coupled simulation reduced surface melting (Fig. 4f) and calving rates (Fig. 4h) outweigh reduced accumulation rates, and hence the freshwater-induced surface cooling (Fig. 5c) provides a net-negative feedback to ice-sheet melting. As a consequence of the substantial differences in AIS mass balance between SSP5-8.5 and SSP5-8.5_MWOFF, the rate of total ice volume loss and the corresponding rate of SL contribution are decelerated when accounting for the fully coupled system (Fig. 2d, Fig. 3b) and the surface temperature effects (Fig. 2a). Although it has been suggested that AIS meltwater fluxes

could remotely impact the GrIS via changes in Atlantic Meridional Overturning circulation and interhemispheric heat fluxes[46], we do not find any noticeable changes in the GrIS response between SSP5-8.5 and SSP5-8.5_MWOFF. A higher-resolution climate simulation may be required to explain the teleconnection at the end of 21st century shown in Supplementary Fig. S6d.

When hydrofracturing and ice-cliff failure parameterizations are turned off in the additional model experiments (Supplementary Table S1), the AIS meltwater flux still decelerates global warming (experiments SSP5-8.5_HFCMOFF and SSP5-8.5_MWHFCMOFF, Supplementary Fig. S4a orange and blue lines). However, the negative and positive coupled feedbacks on SL rise related to the meltwater flux are more in balance (Supplementary Figs. S4b, S5). Despite meltwater and calving fluxes being substantially reduced relative to SSP5-8.5, the surface cooling is nearly as strong in SSP5-8.5_HFCMOFF, due to the cooling becoming less efficient with increasing meltwater flux amplitude[13]. Without hydrofracturing, however, increased surface temperatures and rainfall do not directly impact the calving flux, therefore surface temperature-related feedbacks are weaker (calving is still stronger in SSP5-8.5_MWHFCMOFF than in SSP5-8.5_HFCMOFF to compensate for changes in other fluxes, in particular reduced basal melting).

## Sensitivity of subsurface Southern Ocean warming

To analyze the sensitivity of the Antarctic ice-shelves to SSO warming, first, we obtained new equilibrium conditions from the 10 member initial conditions by doubling the SSO temperature anomaly (with respect to 1850 CE) near the Antarctic ice-shelves during 650 years without greenhouse forcing. SSO temperatures in the Antarctic ice model $T^{IM}$ are calculated using

$$T^{IM} = 2 \times (T^{LC} - T^{LC}_{1850}) + T^{LC}_{1850} \qquad (1)$$

where $T^{LC}$ is the 400 m ocean temperature simulated in LOVECLIM and $T^{LC}_{1850}$ is the corresponding LOVECLIM temperature in the year 1850. Subsequently, we ran a 10 member ensemble covering the historical

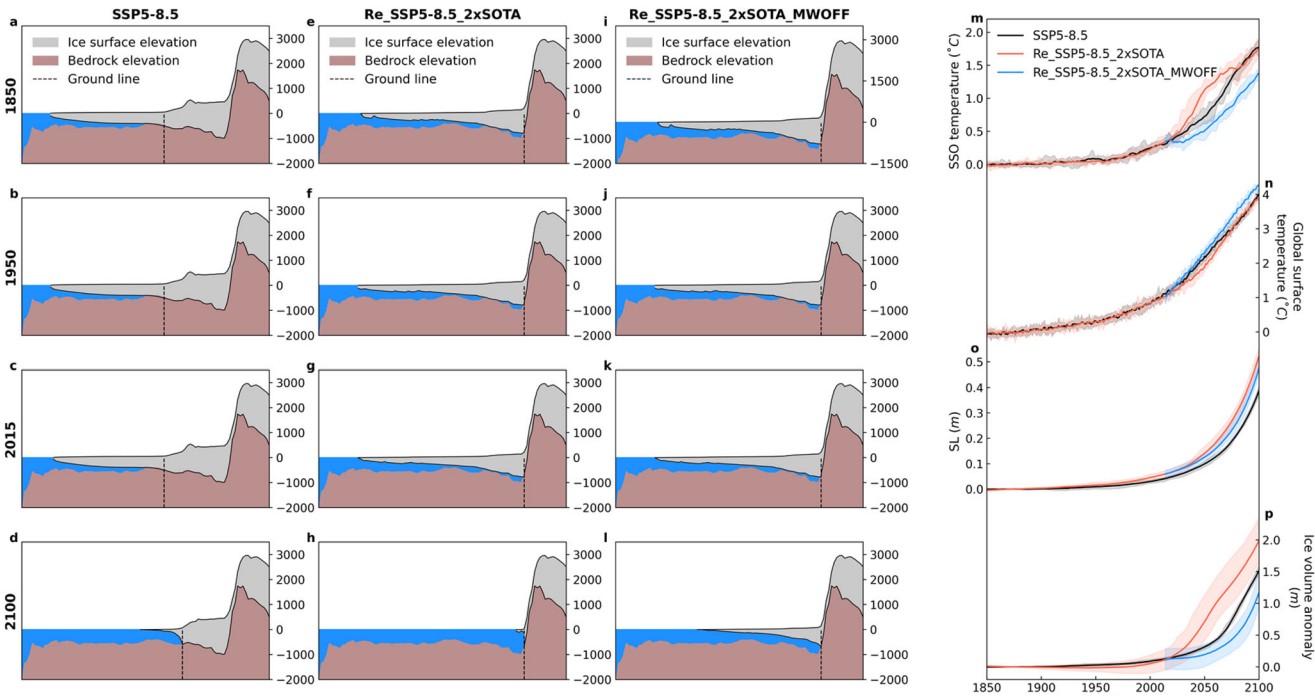

**Fig. 6 | Transections of the Antarctic Ross ice-shelf, and global temperature and SL. a–l** Transects of Antarctic Ross-ice-shelf simulated in (**a–d**) SSP5-8.5, (**e–h**) Re_SSP5-8.5_2xSOTA and (**i–l**) Re_SSP5-8.5_2xSOTA_MWOFF experiments in 1850, 1950, 2015 and 2100 CE. Dashed lines indicate grounding lines. **m–p** Time series of annual anomalies (relative to the 1850–1900 CE mean) of (**m**) subsurface Southern Ocean (SSO) temperature, (**n**) global surface temperature, (**o**) sea-level (SL) and (**p**) Antarctic ice-sheet (AIS) net mass balance in sea-level-equivalent (SLE). Different colors represent the SSP5-8.5 with historical (black line), Re_SSP5-8.5_2xSOTA (red line) and Re_SSP5-8.5_2xSOTA_MWOFF (blue line) simulations. Horizontal scales of **a–l** are shown in Supplementary Fig. S9 as a red line.

period and the SSP5-8.5 scenario with/without Antarctic meltwater flux (experiments Re_SSP5-8.5_2xSOTA and Re_SSP5-8.5_2xSOTA_MWOFF). Warming SSO temperature (Fig. 6m) increases basal melting under the Antarctic ice-shelves, thereby accelerating grounding line retreat (Fig. 6e–h) relative to SSP5-8.5 (Fig. 6a–d). This is most evident in Ross ice-shelf which vanishes completely by 2100 CE, leading to an integrated freshwater input of 2 ± 0.35 m SLE total AIS mass by 2100 CE (Fig. 6p) and de facto SL rise of 0.5 ± 0.04 m (Fig. 6o). Ice calving is the largest term in the mass balance over AIS (Fig. 4e–h). However, as the shelf retreat accelerates mainly due to the basal melting, the role of ice calving term diminishes (Supplementary Fig. S6g, h).

The result of increased Antarctic meltwater fluxes by enhanced SSO warming concurs with our previous discussion of a global warming slowdown by 0.4 °C, relative to a simulation without such coupling (Fig. 6n red and blue lines). However, this sensitivity experiment increases SL by an additional 3 cm SL rise (Fig. 6o). Not unexpectedly, SSO warming does not show any significant influence on the GrIS (Supplementary Fig. S6a–d).

## Discussion

Here we used the coupled three-dimensional climate-ice-sheet model LOVECLIP to better understand the impact of ice-sheet/ice-shelf/ocean/atmosphere coupling processes on the future evolution of GrIS and AIS, and to estimate their respective contributions to SL rise.

In our high-end emission scenario, the GrIS and AIS each contribute about 60–70 cm to global mean SL rise over the next 130 years. Even though for SSP2-4.5 and SSP5-8.5 global surface temperatures are projected to increase at a reduced rate after 2100 CE (Fig. 2e), the ice-sheet contributions to SL continue to accelerate beyond 2100 CE (Fig. 2g, h), mostly driven by accelerated surface melting in case of the GrIS and due to a combination of effects for the AIS. According to our simulations, limiting 21st century global surface temperature rise to 2 °C above the pre-industrial level[41,47] would be insufficient to

slowdown the rate of global SL rise over the next 130 years[42]. Only the more aggressive low greenhouse gas emission scenario (SSP1-1.9), with temperatures leveling off below 1.5 °C (Fig. 2a), avoids SL rise acceleration (Fig. 2f). A longer-term warming and SL perspective until 2500 CE (Supplementary Table S1) illustrates that for the SSP2-4.5 scenario, SL rise due to GrIS melting accelerates for over 250 years after maximum global warming rates occur, and peaks at over 0.3 cm/year shortly before 2300 CE (Supplementary Fig. S7g). The Antarctic SL contribution for SSP2-4.5 fluctuates between 0.2 and 0.3 cm/year from 2200–2500 CE (Supplementary Fig. S7h). This indicates an even more prolonged response and larger commitment to SL rise due to 21st century warming, with the total AIS contribution reaching about 1.1 m by 2500 CE. In contrast, the aggressive greenhouse gas reduction scenario SSP1-1.9 with temperatures leveling off at less than 1.5 °C (Supplementary Fig. S7a, e) is sufficient to prevent substantial ice-loss in Antarctica (Supplementary Fig. S7h) over the next centuries.

Our results are to some extent consistent with recent uncoupled single-hemisphere ice-sheet model simulations[5,8,38] which also show the tendency for unabated SL acceleration over the next two centuries in response to strong greenhouse gas forcing. One of the key advantages of our coupled model setup, even though it uses lower oceanic resolution, is that it allows us to quantify the role of meltwater forcing and calving fluxes on the stability of the ice-sheet. This is particularly timely because recent observations already show a marked increase in AIS meltwater fluxes[14,15], which could increase SO stratification, reduce vertical heat exchange, increase subsurface temperatures and lead to enhanced basal melting. Numerous studies have suggested that, in combination with MISI and MICI, this may set in motion a run-away effect for ice loss. We can clearly see the positive feedback between ice-sheet melting, warm CDW intrusions and basal melting in our model (Figs. 4g, 5b). According to our study, however, the impact of MICI shows a weaker influence on the future SL contributions from AIS, as compared to earlier studies[5,10] (Supplementary Fig. S4).

In our high-emission scenario model simulations that include parameterizations for hydrofracturing, ice-cliff instabilities, and capture sea-ice and atmospheric responses, the net impact of ice-sheet/climate feedbacks on SL rise is negative. These processes strongly contribute to the fast AIS response to warming temperatures, slightly less so in the coupled model than in meltwater-decoupled sensitivity experiments due to reduced surface temperature warming, rainfall, and surface melt. Given this sensitivity, it is plausible that the net effect of including the coupling and even its sign is strongly dependent by models: negative feedbacks related to meltwater fluxes are reduced when hydrofracturing and ice-cliff failure parameterizations are turned off in the model in our sensitivity experiments, where positive and negative feedbacks nearly balance. Furthermore, the net effect of including meltwater coupling is positive in the experiments with increased SSO warming. A more dramatic fall in SL, 22 mm by 2100 in the SSP5-8.5 scenario, was found in the U.K. Earth System Model coupled to BISICLES dynamic ice-sheet model[25]. Thereby, standalone ice-sheet simulations may create misleading projections about the GrIS and AIS contributions to global SL rise. Coupled simulations document that trajectories of future climate change and SL rise depend on the complex and delicate balance of climate-ice-sheet coupling processes, some of which are not yet well constrained observationally. The results presented here from our earth system model of intermediate complexity multi-parameter simulations may provide further guidance in designing new coupled climate-ice-sheet model simulations using the next generation of coupled general circulation models.

Whereas the LOVECLIP model employed here captures climate-ice-sheet coupling in both hemispheres, as well as iceberg routing and thermodynamic effects of melting icebergs, it does not resolve sub-shelf ocean dynamical processes[48,49], interactive changes in ocean bathymetry, as well as narrow coastal currents, which can play an important role in basal melting[50]. Moreover, several important ice-sheet processes are still not well constrained, and both the ice-sheet/ice-shelf model and the atmospheric model use relatively low horizontal resolutions of ~20 km in Antarctica and ~5.6°, respectively. This can lead to additional biases in ice-stream dynamics and poleward moisture transport, respectively. Additionally, glacial-isostatic adjustment[51] other than vertical bedrock response by elastic lithosphere is not fully implemented in LOVECLIP. Nevertheless, our simulations also illustrate that ice-sheet ocean/atmosphere coupling, which can account for individual mass balance differences of 0.5-0.8 cm/year SLE over the next ~100 years (Fig. 4f–h), is a first order process that needs to be included in future SL assessments.

## Methods
The coupled climate-ice-sheet-ice-shelf-iceberg model LOVECLIP[26] is based on the earth-system climate model LOVECLIM[27] and the Penn State University Ice-Sheet/Shelf model (PSUIM)[5,28,29]. Here we review its main components, the coupling algorithm, experimental setup, and highlight modeling differences to previous studies.

### LOVECLIM earth system model
The earth system model of intermediate complexity[27] used here, LOVECLIM model (version 1.3), includes ocean, atmosphere, sea-ice, iceberg components as well as a vegetation model. The atmospheric model ECBILT[52] uses a T21 spectral truncation (corresponding to ~5.6° × 5.6° horizontal resolution) and the prognostic quasi-geostrophic atmospheric equations are solved on three vertical levels. The model includes parameterizations of ageostrophic terms[53] to better capture tropical dynamics. The free-surface primitive equation ocean model, CLIO[54–56], which is also coupled to a thermodynamic-dynamic sea-ice model, adopts a 3° × 3° horizontal resolution and 20 vertical levels. The coupling between atmosphere and ocean is expressed in terms of freshwater, momentum and heat

flux exchanges. An iceberg model integrates iceberg trajectories, melting and freshwater release along individual simulated iceberg trajectories[57,58]. VECODE[59], the terrestrial vegetation model of LOVE-CLIP calculates the temporal evolution of annual mean desert, tree and grass fractions in each land grid cell. LOVECLIM has been used extensively to study the earth system response to a variety of boundary conditions[60–63]. Here the model is configured with a present-day land mask and an open Bering Strait.

### PSUIM ice-sheet/shelf model
The ice-sheet/shelf model PSUIM[5,28,29] is used here in a bi-hemispheric configuration. By adopting shallow ice and shallow shelf approximations, the model retains the ability to simulate streaming and stretching flow and to capture ice streams and floating ice-shelves. Floating ice-shelves, grounding line migration, and basal ice fluxes are parameterized[21]. PSUIM estimates the surface energy and ice mass balance by accounting for contributions from changes in temperature and radiation[64,65]. Similar to previous versions of the model[5,29], we include parameterizations for enhanced calving caused by rainwater-driven hydrofracturing and surface melting, as well as a representation of marine ice-cliff failure. Through calculating changes of ice calving, floating-ice, grounding line migration and pinning by bathymetric bedrock perturbations, the SL is estimated. A horizontal resolution of 1° latitude and 0.5° longitude is used in the Northern Hemisphere and a stereographic grid is adopted for Antarctica with a resolution of 20 km.

The model version employed here differs from the one used in a recent SL study[5] in that our spatial resolution is lower over Antarctica. Moreover, different parameters were used, namely those characterizing sub-ice-shelf ocean melting (OCFAC), the coefficient in the parameterization of hydrofracturing due to surface liquid (CREVLIQ), and the maximum rate of horizontal wastage due to ice-cliff failure (VCLIF). The default parameter values are OCFAC = 1.0, CREVLIQ = 100 m per $(m/year)^{-2}$ and VCLIF = 3 km/year[5]. Here we use a value of OCFAC = 1.5, which were chosen such that the AIS has a realistic extent under pre-industrial conditions and for the corresponding LOVECLIM climate forcing. Additionally, LOVECLIP realistically simulates the Antarctic ice velocity and shape of ice-shelves compared to the mean of 1996–2016 Antarctic ice velocity obtained from the satellite data[66,67] (Supplementary Fig. S11). Although, the size of the ice-shelves and the associated outflow velocities are overestimated.

### Climate-ice-sheet model coupling (LOVECLIP)
A coupling algorithm exchanges variables and boundary conditions between LOVECLIM and PSUIM in both hemispheres (Supplementary Fig. S1), in a series of alternating climate and ice-sheet model runs ("chunks")[26,68–70]. The chunk length is set to 1 year for LOVECLIM and PSUIM. The ice model is forced by monthly LOVECLIM surface air temperature, precipitation, evaporation, solar radiation and annual mean subsurface ocean temperature. LOVECLIM has polar temperature and precipitation biases, similar to those documented for more complex CMIP5 models[71]. Present-day climatological surface air temperature and precipitation biases, as well as subsurface ocean temperature biases near Antarctica, are removed in the coupler through a bias correction[70]. Surface air temperature and precipitation are downscaled vertically to the PSUIM grid with applied lapse rate corrections[26,28]. Subsurface ocean temperature is interpolated under ice-shelves on the PSUIM grid.

LOVECLIM's surface land-ice cover and orography are updated using the simulated ice-sheet and vertical bedrock evolution from PSUIM which is based on elastic lithosphere response with fixed bedrock response time[28]. The spatial distribution of liquid runoff into the ocean is calculated based on model topography and the calving flux is released as icebergs into the LOVECLIM iceberg model. Both liquid runoff from the surface and basal melting are released into the surface ocean. Ocean currents and wind-drag subsequently steer the icebergs

and along their pathways they melt and cool the ocean. Contributions to SL changes are calculated in PSUIM in both hemispheres, and take into account the bedrock response[28]. The total SL evolution is calculated in the coupler based on the Northern Hemisphere and Antarctic contributions.

## Spin-up and initial conditions

The model is initialized from constant pre-industrial conditions. Because of the different equilibration timescales of the ice-sheet and climate components, asynchronous coupling is used to obtain equilibrated initial conditions. In particular, the model has been integrated for 120 chunk lengths with chunk lengths of 50 years for PSUIM and 5 years for LOVECLIM, and then 2000 chunk lengths with chunk length of 1 year for each ice model and LOVECLIM, which results in 8000 years of spin-up for the ice component and 2600 years for LOVECLIM (Supplementary Table S1), after climate trends are negligible (Supplementary Fig. S10). Ensembles of 10 members with different initial conditions have been conducted for each of the experiments, with the initial conditions taken from the last 100 chunks of the spin-up run.

## Statistics of forced and unforced ice-sheet mass balance

To compare whether the observed ice-sheet mass balance estimated from the Gravity Recovery and Climate Experiment (GRACE)[33] for the period 2002–2020 CE is consistent with the null hypothesis of unforced ice-sheet variability, we conducted a 5000 years LOVECLIP control experiment with constant, pre-industrial $CO_2$ concentrations (CTR) (Supplementary Table S1). There is still a remaining very weak drift in the unforced simulation, which amounts to −1.5 cm for GrIS and −4 cm for AIS of mass balance in SLE over 5000 years (Supplementary Fig. S2). The drift is removed by high-pass filtering over than 80 years, the net mass balance and the resulting high-frequency components are then used as an estimate for the unforced ice-sheet variability. Each 19-year chunk is cut after high-pass filtering to get 19-year trends of natural variability (Fig. 1).

## Scenario simulations

To assess the ice-sheet sensitivity to different greenhouse gas emission pathways, we conducted a suite of coupled scenario simulations (each with 10 individual ensemble members), in which $CO_2$ concentrations in LOVECLIP follow the historical from the year 1850 to 2014, and SSP1-1.9, SSP2-4.5 and SSP5-8.5 from the year 2015 to 2150 (Supplementary Table S1). In terms of the Antarctic contribution to SL by 2100 CE the LOVECLIP model projections (Fig. 2) lie within the range simulated by offline models which were forced by RCP2.6, 4.5 and 8.5 climate scenarios[6,39,40]. For the Greenland ice-sheet we find a similar agreement with other modeling studies[6,36,37] and the "likely range" of provided by the 5[th] assessment report, WG1 of the Intergovernmental Panel on Climate Change (Chapter 13)[4]. The SSP1-1.9 and SSP2-4.5 simulations were further extended until 2500 CE (Supplementary Fig. S7).

To further quantify the impact of climate-ice-sheet coupling in the Southern Hemisphere in global SL rise we conducted an additional SSP5-8.5 sensitivity experiment, for which the AIS liquid runoff and iceberg calving balance net precipitation over Antarctica (experiment SSP5-8.5_MWOFF) (Fig. 5 and Supplementary Table S1). To explore the impact of AIS hydrofracturing and ice-cliff failure parameterizations, we also obtained ensembles with these parameterizations turned off (CREVLIQ = 0 m per (m/year)$^{-2}$ and VCLIF = 0 km/year) with and without meltwater flux coupling (experiments SSP5-8.5_HFCMOFF and SSP5-8.5_MWHFCMOFF) (Supplementary Figs. S4, S5).

Another sensitivity experiment of the Antarctic ice-shelves to SSO warming is conducted by doubling the SSO temperature anomaly (relative to 1850 CE) to the Antarctic ice shelves in the SSP5-8.5 scenario with/without Antarctic meltwater flux (experiments Re_SSP5-8.5_2xSOTA and Re_SSP5-8.5_2xSOTA_MWOFF). Specifically, SSO temperatures in the Antarctic ice model $T^{TM}$ are calculated using Eq. (1).

Applying doubled SSO temperature anomaly would lead to a different climate equilibrium. Therefore, new initial conditions were created. To get the new equilibrium conditions we added the SSO temperature anomalies from the 10 member initial conditions for 650 years, with fixed pre-industrial $CO_2$ concentration.

## Data availability

The data of the LOVECLIP model simulations presented in this study is available as NetCDF files on the ICCP climate data server at https://climatedata.ibs.re.kr. The data server allows access to the data either through the Live Access Server (LAS) web interface or as OPeNDAP data which can be directly used in client software such as matlab, ferret, panoply, python, etc.

## Code availability

The LOVECLIM mode code is available on https://www.elic.ucl.ac.be/modx/index.php?id=289. Please contact Dr. David Pollard for the PSUIM model code.

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

## Acknowledgements

J.-Y.P., A.T., J.-Y.L., D.C., A.B.N. were supported by the Institute for Basic Science (IBS), Republic of Korea, under IBS-R028-D1. F.S. was supported by NSF grant 1903197 and NASA grant 80NSSC20K1241. The LOVECLIP simulations were conducted on the IBS/ICCP supercomputer *"Aleph"*, 1.43 peta flops high-performance Cray XC50-LC Skylake computing system with 18,720 processor cores, 9.59 PB storage, and 43 PB tape archive space and supercomputer Cheyenne (doi:10.5065/D6RX99HX) provided by NCAR's Computational and Information Systems Laboratory, sponsored by the National Science Foundation. The team further acknowledges contributions from Dr. Malte Heinemann, who helped with the coupling of an earlier version of the model and Dr. David Pollard for making the Penn State ice-sheet model available.

## Author contributions

J.-Y.P., F.S., A.T and J.-Y.L. designed the study. J.-Y.P. and F.S. conducted the simulations, wrote the initial manuscript draft, and produced all figures. J.-Y.P., F.S., A.T., D.C., J.-Y.L. and A.B.N. contributed to the interpretation of the results and to the improvement of the manuscript.

## Competing interests

The authors declare no competing interests.
