## [Peer Review File · Nature Communications]

Future sea-level projections with a coupled atmosphere-ocean-ice-sheet modelREVIEWER COMMENTS

Reviewer #1 (Remarks to the Author):

The manuscript of Park and others presents results of a climate model of intermediate complexity (EMIC) where the climate model interacts with both the Greenland Ice Sheet and Antarctic Ice Sheet. Compared to former similar studies (Goelzer et al. 2010), they use the Penn State University Ice Sheet/Shelf model (PSUIM), which allows studying the impact of various mechanisms determining, in particular, the ice loss of the Antarctic Ice Sheet, such as the hydrofracturing, Marine Ice-Sheet Instability (MISI), and Marine Ice Cliff Instability (MICI). Since MICI is hotly-debated Marine Ice Cliff Instability (MICI), e.g., (Pollard, Deconto, and Alley 2015; Edwards et al. 2019; Bassis et al. 2021), this study allows determining its impact in an interacting climate context.

To explore the potential sea-level contribution from Greenland and Antarctica, the authors create an ensemble of 10 simulation members for various future climate scenarios ranging from SSP1-1.9, which is a strong climate mitigation scenario, to the high-end scenario SPP5-8.5. The exploitation of the ensemble enables the authors to present the variations due to differences in the initial climate model conditions. More importantly, they perform an extensive set of parameter studies to assess the impact of parametric choices in determining the sensitivity of the ice sheets' sea-level contribution. They analyze the sensitivity of hydrofracturing, MISI, MICI, and an amplified subsurface ocean warming. The results show clearly that the full interaction between ice sheets and the common climate system delays global warming. Furthermore, although the debated MICI influences the results in the coupled simulations, it is not decisive. Ultimately, the authors reveal that only the most substantial climate mitigation scenario SPP1-1.9 avoids a long-term sustained sea-level contribution from Greenland and Antarctica.

This study is highly relevant for several reasons.

1. The missing feedback loop in so-called standalone ice sheet simulations may create misleading projections about the sea-level contribution of Greenland Antarctica, which has numerous implications for the political and social spheres.
2. Various ongoing activities of coupling ice sheet models into state-of-the-art climate models will benefit from the knowledge gained by studies utilizing EMIC because extensive parameter studies, as presented here, are computational not feasible for these models.
3. This study does not settle the debate about MICI. Still, it may lead to a different angle of the discussion because the inclusion/omission has a relatively low influence in the presented simulations compared to other uncertainties.

It was a pleasure to read this excellent manuscript. The study is well organized and written. The figures are generally of high quality, necessary, and informative.

I recommend the publication of the manuscript after a few minor corrections.

General comments

The initial conditions of the ice sheet state strongly determine their mid-term (century) evolution. Therefore, I understand that the authors have utilized an a-synchronous coupling between the climate model and the ice sheet during the spin-up to cover 8000 years for each ice sheet. Considering the timing of glacial-interglacial cycles (120000 years) or the time since the last glacial maximum (LGM) of about 21000 years before the present, this ice sheet spin-up seems relatively short. Would you please add some more information about the spin-up of the ice sheet into the Method section, such as the initial temperature field and the used starting geometry? In addition, I would like to ask how your results would differ if you would let your model system run an entire glacial-interglacial cycle to spin up the ice sheets?

For Antarctica and, in particular, Greenland, the surface mass balance is central to the evolution of these ice sheets because it determines where and how much ice is essential

gained or lost at the ice sheet-atmosphere interface. Your atmospheric model has a relatively coarse resolution, while the current ablation zone ("melting zone") has a width of several 10 km. Could you please share some information about the surface mass balance pattern that the Greenland ice sheet model receives? How does your downscaling work? You may cite explicitly relevant publications in the Method section to help the reader.

The current partitioning of Antarctica's ice loss ranges from 1:1 to 2:1 for basal melting of ice shelves and iceberg calving (Depoorter et al. 2013; Liu et al. 2015; Rignot et al. 2013). Since iceberg calving contribution seems to be much higher in your simulations than observational estimates suggest, do you have an explanation for the shift towards calving or why the basal melting is too low? Would an artificially amplified basal melting and damped calving be possible to reproduce observational partitioning? If it would be possible, how would it impact the results if you would run your "retune" model system accordingly?

Specific comments

=====

Main document

Page 2, Line 24: I guess your mean: "Ocean-ice-shelf coupling enhances ..."

Page 4, Line 74-88: Here, you talk about the climate fields received by the ice sheets. Would it be possible to state here that you exploit an anomaly coupling approach to deal with the unavoidable biases in the climate state coming from LOVECLIM?

Page 5, Line 94-111: The comparison between GRACE and your simulated total mass balance evolution is an excellent validation. However, after reading this paragraph for the first time, I initially had difficulties understanding the difference leading to the Figure 1 and the Supplementary Figure S3. Could you please clarify this part?

Page 7, Line 137-140: I understand that "retuning" parameters are a common approach to ensure a reasonable representation of the conditions obtained from observational estimates. How strong is the effect if you would have used the former set of parameters or if you would have used the former standalone setup of PSUIM?

Page 7, Line 178-181: Is it possible to provide separately the MISI, hydrofracturing, and MICI contribution spatially integrated across Antarctica? If so, would you please be so kind as to add such a figure to the Supplement?

Page 10, Line 226-229: Various processes are discussed, conveying the information about meltwater release around Antarctica to the Northern hemisphere. Is the not detected teleconnection a specific feature of your model system? Causes the coarse resolution in the ocean model component the hardly noticed teleconnection, although Figure S6d indicates some connection at the end of the simulations?

Page 12, Line 240-244: This result again highlights the importance of the coupling allowing the evolution of feedback. As already mentioned in the general section, I would like to ask, why does calving compensate for other fluxes? Is this compensation an artifact?

Figure 1, caption: Do I understand you correctly: "... (a) histogram of 19-years trends (2002-2020) of Greenland mass balance in CTR"?

Figure 3, c-e: Do you simulate ice on Island, Elsmere Island, for instance? What is the domain covering the Northern hemisphere.

Figure 3, g-h: If I understand it correctly, the black line represents the grounding line position. Is it correct that this line has disappeared in part of the Ross Ice Shelf in

subfigure h? Would it be possible to draw a (dashed) line following the outer ice sheet/shelf edge in the anomaly plots g-h? Since I find it hard to identify a disintegrating ice sheet.

Figure 5, d: You might have missed the subplot label "d"?

Figure 5, caption: You may want to rephrase: "..., (a) the SO subsurface temperature at 400 m depth, (b) ..."?

Figure 6, a-l: Does the vertical black line represents the grounding line position? Please mention in the caption what the vertical black line represents. Since I can hardly identify the ice sheet/shelf shape in my version, would you (a) restrict the depth to -2000 m, for example, (b) change the color of the ice or surround the ice shape with a line? If you do not like to change the color of the ice, you may mark the frontal position with a "*".

Method

Page 29, Line 553-554: Is the resolution of 20 km sufficient to represent the grounding line migration reasonably?

Page 30, Line 575-577: Is the meltwater from the ice surface released into the surface ocean? Is the basal melting of ice shelves also released into the surface ocean? Please clarify these points.

Page 30, Line 579-580: Does PSUIM consider the bedrock's elastic and viscose response? Does PSUIM consider differences in the response time of the bedrock topography to load changes between East and West Antarctica? Please clarify.

Page 30, Line 586: Do you mean: "with chunk lengths of 50 years for the ice sheet model and 5 ..."

Page 31, Line 588-589: Can you provide some information about the internal ice-sheet temperature at the start of the scenarios simulations?

Supplement Material

Figure S4, e-f: You may have used some wrong subfigure labels. Please check.

Figure S4, f-g: The sudden drop in the parameter study simulations is probably a stock due to the changed conditions. If so, please mention it in the figure caption. If it is not the case, could you please explain it?

Figure S6, d: Do you know what drives the 50% enhanced calving in Greenland for the amplified ocean temperature around Antarctica at the end of the simulation, while the basal melting in Greenland is nearly identical?

Figure S7, h: Are the fluctuations in Antarctica's sea-level contribution for SSP2-4.5 driven by waxing and waning ice shelves and subsequent calving events?

Supplement table: You may add to the table caption the information that you have computed ten ensemble members for each climate scenario.

Bibliography

Bassis, J. N., B. Berg, A. J. Crawford, and D. I. Benn. 2021. "Transition to Marine Ice Cliff Instability Controlled by Ice Thickness Gradients and Velocity." *Science* 372 (6548): 1342-44. <https://doi.org/10.1126/science.abf6271>.

Depoorter, M.A, J.L. Bamber, J.A. Griggs, J.T.M. Lenaerts, S.R.M. Ligtenberg, M.R. van den Broeke, and G. Moholdt. 2013. "Calving Fluxes and Basal Melt Rates of Antarctic Ice Shelves." *Nature* 502 (7469): 89-92. <https://doi.org/10.1038/nature12567>.

Edwards, Tamsin L., Mark A. Brandon, Gael Durand, Neil R Edwards, Nicholas R. Golledge, Philip B Holden, Isabel J. Nias, Antony J. Payne, Catherine Ritz, and Andreas Wernecke. 2019. "Revisiting Antarctic Ice Loss Due to Marine Ice-Cliff Instability." *Nature* 566 (7742): 58–64. <https://doi.org/10.1038/s41586-019-0901-4>.

Goelzer, H., P. Huybrechts, M.-F. Loutre, H. Goosse, T. Fichefet, and A. Mouchet. 2010. "Impact of Greenland and Antarctic Ice Sheet Interactions on Climate Sensitivity." *Climate Dynamics* 37 (5–6): 1005–18. <https://doi.org/10.1007/s00382-010-0885-0>.

Liu, Yan, John C Moore, Xiao Cheng, Rupert M Gladstone, Jeremy N Bassis, Hongxing Liu, Jiahong Wen, and Fengming Hui. 2015. "Ocean-Driven Thinning Enhances Iceberg Calving and Retreat of Antarctic Ice Shelves." *Proceedings of the National Academy of Sciences* 112 (11): 3263–68. <https://doi.org/10.1073/pnas.1415137112>.

Pollard, David, Robert M Deconto, and Richard B Alley. 2015. "Potential Antarctic Ice Sheet Retreat Driven by Hydrofracturing and Ice Cliff Failure." *Earth and Planetary Science Letters* 412: 112–21. <https://doi.org/10.1016/j.epsl.2014.12.035>.

Rignot, E., S. Jacobs, J. Mouginot, and B. Scheuchl. 2013. "Ice-Shelf Melting Around Antarctica." *Science* 341 (6143): 266–70. <https://doi.org/10.1126/science.1235798>.

Reviewer #2 (Remarks to the Author):

Park et al Future sea-level projections with a fully coupled atmosphere-ocean-ice-sheet model

This paper presents a new set of model simulations for future sea level change under three greenhouse gas emissions scenarios, using a method that includes full ice-ocean-atmosphere coupling. The simulations run to 2150, and are able to capture feedbacks between different model components. The experiments show that combined Greenland and Antarctic ice sheet contributions to sea level by 2150 could reach 1.4 m under a high emissions scenario, or as little as 0.2 m under the most stringent mitigation scenario. The results also suggest that the rate of sea level contribution declines through time as a consequence of atmospheric cooling arising from meltwater feedbacks, compared to uncoupled simulations. The authors conclude that more advanced models, such as theirs, are necessary for realistic estimates of future sea-level rise.

The manuscript is well written and the figures are clear. The Supplementary Information file also provides additional useful figures and data tables. Overall I found the paper interesting and the experiments well formulated, and the results will certainly be of relevance to the wider community. There are only two key points that I feel should be addressed in a revised version.

1) Throughout the manuscript the authors assert that their full coupling is new and that their study is the first to account for the atmospheric cooling effect exerted by ice sheet meltwater (e.g. I26, I213-4, I222, I320). However, whilst elements of their model design are indeed distinct from what has been undertaken previously, it seems odd to me that, despite being cited once or twice in other contexts, the authors don't acknowledge the simulations we presented in Golledge et al., 2019 which specifically DO incorporate the effects of MW-induced atmospheric cooling on ice sheet evolution, both in Greenland and Antarctica. Ours was an asynchronous rather than fully-coupled set up, but nonetheless, our simulations included atmospheric as well as oceanic temperature perturbations arising from meltwater discharge from the ice sheets:

G19 p67. "In the Southern Hemisphere, widespread air surface cooling of more than 2 °C and local cooling of up to 4 °C is evident south of approximately 40° S (Extended

Data Fig. 1b)."

G19 p71. "by incorporating ice–ocean–atmosphere feedbacks, our simulations predict substantial atmospheric cooling over mid- to high southern latitudes (Extended Data Fig. 1b), which will offset approximately 0.5–3.5 °C of the atmospheric warming predicted by CMIP5 models (Extended Data Fig. 2c, d) and thus reduce melt."

Because we incorporated the effects of this atmospheric cooling into subsequent runs with the ice sheet model, we captured the consequences of this feedback in a way that eg Bronselaer et al 2018 had talked about but hadn't been able to do (they ran the GFDL atmos-ocean model but had no dynamic ice sheet component). And of course, as the authors are well aware, much of this simply builds on experiments from Menviel et al 2010, Golledge et al 2014 and others. I think also the most recent DeConto and Pollard simulations (DeConto et al 2021) also include atmospheric feedbacks from freshening:

D21 p88. "Including meltwater discharge in CESM expands Southern Ocean sea ice, stratifies the upper ocean, and warms the subsurface (400 m water depth) by 2–4 °C around most of the Antarctic margin in the early 22nd century⁴⁶. Conversely, expanded sea ice suppresses surface atmospheric warming by more than 5 °C, slowing the onset of surface melt and hydrofracturing in the ice-sheet model. The net result of competing sub-surface ocean warming (enhanced sub-shelf melt) and atmospheric cooling (reduced surface melt) produces a substantial negative feedback on the pace of ice-sheet retreat (Fig. 1h). "

So I think it might be worth revisiting some of the wording in the current manuscript to avoid claiming novelty for this particular aspect.

2) The simulations presented use the LOVECLIM intermediate complexity model, which is a low resolution (3x3 deg) simplified atmos-ocean model. It is widely used for both palaeoclimate and future simulations, and does a reasonably job of capturing first order global responses. However, as the authors note (l132), the model has a relatively low temperature sensitivity, which may affect their results. Although they acknowledge this, I think it might be also worth highlighting that the sensitivity isn't globally uniform, and in the ocean it is especially problematic. For example, under certain circumstances LOVECLIM may simulate a global mean SST warming that is similar to CMIP-class models, but the latitudinal gradients are very different. Typically, LOVECLIM vastly underestimates polar amplification, especially in the Arctic, and often even fails to capture mid or low-latitude SST anomalies. The atmospheric response is somewhat better, but is still too cold pretty much everywhere. It might be useful, therefore, to mention this in the text, and perhaps show a figure that compares the future climate states from LOVECLIM (without the meltwater effect) alongside the comparable CMIP ensemble mean for the same scenario.

On the whole though it's a nice paper and an important contribution to the literature.

Nick Golledge
30th May 2022

Reviewer #3 (Remarks to the Author):

Summary

Park et al have conducted simulations of future climate change using an intermediate complexity Earth System model that includes an ice sheet model component. Using this,

they present projections of the contributions of mass loss from the Greenland and Antarctic ice sheets to global mean sea level rise under several different scenarios, with a focus on the period 2015-2150. They also look at some of the impact of omitting aspects of their parameterisation of the coupling between the Southern Ocean and Antarctic ice sheet. Although there are some novel aspects to Park et al.'s approach, their projections are based on a model with little explicit representation of many important physical processes and they show little to convince a skeptical reader that its performance is adequate in a number of crucial areas. There are many aspects of the behaviour of continental ice sheets and their interactions with the climate system that are poorly understood and practically difficult to model realistically; consequently the community's best efforts at making near-term quantitative projections of ice mass loss under future climate change come with extremely large uncertainties (Goelzer et al. 2021, Seroussi et al. 2021). Unfortunately, Park et al.'s simplistic experiment design hardly considers the manifest uncertainty present in our knowledge of the climate-ice system in general and their model in particular. Given the current state of the science in this area, unfortunately I do not think that this study presents projections that can be considered robust or that the analysis they present adds much to our understanding of the relevant parts of the climate system.

General Comments

Understanding and projecting the future ice sheet contribution to sea level rise is a topic of great importance that has seen much progress in recent years via advances in numerical modelling, most prominently perhaps with ISMIP6, the international multi-model community effort to simulate the 21st century evolution of ice sheets consistent with the recent CMIP6 climate projections. If nothing else, the results of ISMIP6 highlight that it is very difficult to robustly model the recent past and near future of the Greenland and Antarctic ice sheets, even with state of the art climate and ice sheet models, and that various groups' attempts to do so lead to a very wide range of possible results. Few of the climate modelling groups that participated in CMIP6 can use explicit coupling to a ice sheet models in their simulations because of the many technical and scientific challenges to doing so plausibly, although two did (CESM2 (Muntejewerf et al. '20) and UKESM1 (Siahaan et al '21) who report on aspects of next-century simulations coupled to the Greenland and Antarctic ice sheets respectively). In this paper, Park et al. use a model with (when compared to CMIP6 climate models) much lower resolution and low process complexity. Their model includes coupling to an ice sheet component, although their computational shortcuts require the links between climate and the ice sheet model to be heavily parameterised and bias corrected. With this, they have run a small (10 member) initial-condition ensemble of climate change and ice sheet projections and reported on the ensemble average global temperature change and ice sheet mass loss contributions to GMSLR for the same sort of timescale as ISMIP6 did. I'm afraid I simply do not think that decadal-scale projections of the centre-of-probability-distribution global mean sea level rise is a plausible use of this class of model, and Park et al. have not convinced me with this paper.

As alluded to above, the production of climatic boundary conditions for ice sheets is particularly problematic for even state of the art global ESMs. Regional detail in many aspects of both the atmosphere and oceans around the ice is crucial to simulating the evolution of the continental-scale ice sheet. Changes in high latitude atmospheric humidity and circulation, changes in the partitioning of precipitation between snow and rain, the density contrast between open ocean and water on the continental shelves around Antarctica that is itself controlled by the interplay of westerly winds, coastal currents, sea-ice processes and freshwater from the shelves themselves...the list of processes that need to be modelled in an ESM and fed back on as the ice sheets evolve on this timescale is very extensive, and we do not understand many of them well enough to make this sort of projection reliably even in our most sophisticated models. Many of these things seem to be heavily parameterised in LOVECLIP. Such parameterisations can allow unknown processes to be absorbed into a framework where

variation and sensitivity can be investigated via different tuning values. However, Park et al. have not done such an investigation. Rather they seem to have tuned their web of parameterisations and feedbacks that contain all this crucial detail to one state and simply run a CMIP6-like set of best-estimate future climate change scenarios. In a field where it is apparent that results are very heavily model-dependent this single-tuning approach is not likely to produce a robust set of projection numbers, and the lack of process detail contained in their model means that little physical understanding can be gained from analysis of this single configuration of their model. The authors note this heavy qualitative dependence of results on the model and approach to coupling themselves in their Discussion (line 311), but I don't see that they have attempted to do anything to alleviate this weakness in what they have done or the numbers they report.

There is almost nothing to provide confidence that their tuned configuration actually does a reasonable job of simulating the necessary regional detail or sensitivity of almost any aspect of historical climate change or ice sheet evolution. Showing that their ice sheet-integrated mass loss trend to be within the uncertainty range of one short observational record is just not sufficient. Regional features (eg recent ocean warming and increased ice flow rates in the Amundsen Sea region, or the appropriate rate and spatial pattern of surface mass balance at the margin of Greenland) are crucial for determining the near-term evolution of the ice sheets, and there is no evaluation of such factors at all, even though many of these fields should be explicitly present in LOVECLIP rather than wholly parameterised. Some of the results that are shown lead me to think that their model may not in fact be capable of realistically simulating basic important climate processes: it is concerning that the global temperature sensitivity to CO₂ in this tuning is so low, that there is apparently no sensitivity of their Antarctic precipitation to global warming (cf Vignon et al., 2021) and that there are unexplained "bias corrections" used for many of the fields being coupled. None of this means the model is necessarily wrong, but they are all aspects that make it look like this is not an appropriate model to produce a robust multi-decadal projection of overall sea level rise (with only scenario uncertainty reported), or to trust that the component interactions underlying it are happening in a realistic manner.

For projections of the next century, it is important to have both the current state and centennial trends in different aspects of ice sheet evolution correct, not unlike a weather forecast where the initial conditions of the system matter. Little is said to give confidence that the initial conditions of this coupled system is appropriate, past the evaluation of the twenty year trend for overall ice sheet mass at the end of the century. Appropriate initialisation of coupled climate-ice sheet models is crucial in the projection question, and is an ongoing topic of research. Park et al.'s initialisation strategy is not very clearly laid out, but if it is to aim for reasonably steady ice sheet extents in balance with the preindustrial climate, I would question whether this is an appropriate aim. For the period in question the Earth's ice sheets are still transiently adjusting from the Last Glacial Maximum, so aiming for a balance with preindustrial climate does not seem likely to give rise to an initial state likely to lead to accurate projections, and sampling more members (10 are used) from within this preindustrial simulation doesn't seem likely to cover an appropriate range of possible realistic initial states.

In summary, I think there is not enough shown to have confidence that Park et al. are starting their coupled climate-ice projections from a plausible end of 20C state, nor that the single tuning Park et al have chosen for their model gives rise to an appropriate ice sheet sensitivity to 21C climate change. There is not enough process detail present in their model to learn anything robust about the physical interactions that would happen in reality, and despite some attempt with the briefly-analysed `_MWOFF` and `_HFC` variations, not enough of the model parameter space is explored to give useful quantification of the possible different states of the coupled climate-ice system or strengths of the feedbacks between components that they do have. In my opinion, this is a weak application of an Earth system model of this class. Its strengths are relative computational affordability, which allows for simulations of longer timescales unavailable to more explicit models, or large ensembles that help quantify initial

condition uncertainty or the parametric uncertainty in the various component models or the relative importance of different coupling pathways.

I'm afraid do not think this study should be published as it is, and suggest that the authors should radically overhaul their model evaluation strategy and experiment design if they wish to use LOVECLIP to provide useful information on 21st century sea level rise and climate-ice sheet interactions.

References

Goelzer et al., "The future sea-level contribution of the Greenland ice sheet: a multi-model ensemble study of ISMIP6", TC 2020
Muntjewerf et al., "Greenland Ice Sheet Contribution to 21st Century Sea Level Rise as Simulated by the Coupled CESM2.1-CISM2.1", GRL 2020
Seroussi et al., "ISMIP6 Antarctica: a multi-model ensemble of the Antarctic ice sheet evolution over the 21st century", TC 2020
Siahaan et al., "The Antarctic contribution to 21st century sea-level rise predicted by the UK Earth System Model with an interactive ice sheet ", TCD 2021
Vignon et al., "Present and Future of Rainfall in Antarctica", GRL 2021

Specific comments

line 1: "fully coupled" is an ambiguous and usually inaccurate term for any model of the Earth System, including this one. Many possible component couplings are simplistically parameterised or omitted entirely, and if "fully" is meant simply to indicate that all the components that the modellers wanted to include are permitted to exchange information, it is vague and subjective.

line 19: Although these interactions are undoubtedly important, I do not think that adding additional complexity to our modelling of elements of the Earth System systems that are as yet poorly understood necessarily leads to more /realistic/ estimates

line 23: "next 130" is inaccurate, given the current date

line 23: Given the well-acknowledged uncertainty in many aspects of projecting future sea level, reporting some uncertainty in these numbers and what it represents in the context of the study would in principle be a good idea

line 26: I'm not sure of the exact context being implied in "had not been considered", but an increase in stratification and thus a cooling influence on the local atmosphere from Southern Ocean freshwater input is not a new result (eg Stouffer et al. 2007) and is implicit in some of the boundary conditions taken from CMIP models for the ISMIP (Nowicki et al 2016.) exercise, since the GCMs include some of the effects of the changing climate on the ice sheets

line 60: It reduces the buttressing effect of the ice shelf /on/ the grounded ice

line 66: Given the uncertainties (amongst others) mentioned on line 46- above and gaps in our fundamental knowledge of these processes, I do not think it is true to imply that such models exist, or that the models that do exist can be said to "capture" these interactions. I would also argue that solid earth interactions that may impact the grounding lines of ice shelves, eg Coulon et al 2021 should not be forgotten in this list.

line 75: Very little is made of any variation between individual members or why each was chosen, and no attempt is made to put numbers to the uncertainty they think is represented by how they have configured the ensemble. In a field where uncertainty in the response and low probability-high impact events are increasingly seen as vital, I find choosing to create an ensemble by sampling only across preindustrial climate initial

condition uncertainty and then not talking about the results bizarre.

line 81: I don't think enough has been done within the context of this study to claim that the impact of these enormously uncertain processes have been robustly "quantified"

line 99: what density is being used in this metric?

line 105: whilst of course it's better than nothing, I don't think that comparison with a single ice-sheet averaged number from a twenty year observational period is anywhere near sufficient to demonstrate that the simulation of the current state of each ice sheet is realistic or that their sensitivity to a changing climate over the next 150 years is likely to be plausible. Nor does it seriously support the implied assertion that the range of variability simulated in CTR is thus representative of non-anthropogenic ice sheet variability in the post-LGM Earth System

line 114: it is not to be expected that either the GrIS or AIS mass should be in balance with preindustrial climate, even neglecting ongoing solid-earth adjustments from the Last Glacial Maximum, so some trend is normal (figure S2). But drawing a linear trend line through what appear to be millennial-scale oscillations in each ice sheet seems unfounded, and the two "sudden" drops in AIS mass in the CTR that do not recover are a bit worrying if you want to claim that the system is in some sense "spun-up"

line 117: I'm seriously concerned that no evaluation whatsoever is done of the performance of the climate model through the 20th century part of the simulation. As is acknowledged, the year 2100 warmings for each of these scenarios are significantly lower than many of the CMIP models, but some confidence in the climate response of relevant parts of the system could be found in the historical period. The level of polar amplification of the global average warming could be verified and would be important, as could shifts in the westerlies over the Southern Ocean. How on earth does the 3 level QG atmosphere cope with climate forcings such as aerosols and ozone that are important for clouds, radiation etc over Antarctica both in this period and in the future under some SSPs?

line 124: although warmer temperatures would indeed be expected to increase surface melt on Antarctica, it is also expected to significantly increase the amount of precipitation there (e.g. Vignon et al. 2020) and it is common in many climate models for this increase in accumulation to outweigh the increase in melt for at least some part of the 21st century. I'm concerned that the LOVECLIP projections appear to feature next to no increase in AIS accumulation (eg figure 4) for any of the scenarios and that this part of the mass balance is hardly mentioned. The partitioning of increases in precipitation between snow and rain is crucial too - it is not mentioned how this is handled in the simple 3 level atmosphere or how it changes in the future.

line 171: it's more than "could be important" - from everything that is known the response of the AIS shelves to warming in the Southern Ocean is acutely sensitive to the regional topography, circulation and salinity as well as temperature. Nothing is shown of whether the modern open ocean state or the coupling method reproduces for instance, the currently observed state of the individual embayments that are expected to be crucial for West Antarctic stability, so it's very difficult to have any confidence in the response of the model. The fact that these processes are entirely dealt with in LOVECLIP via extrapolation of bulk ocean temperature is a major feature of a model being used for near-term projections in this way and should not be mentioned lightly in passing as it is here. Simply doubling the temperature sensitivity of the coupling parameterisation as in experiment x2_SOTA does not begin to address the uncertainty in this aspect of their model projection.

line 174: what processes are involved in this disappearance? This seems an incredible amount of ice to lose on an 80-year timescale, and if it is to be a feature of a model simulation we are meant to rely on for projections it needs to be discussed more. Since

LOVECLIP involves somehow-tuned parameterisations of both MISI and the speculative MICI, is to to be assumed that these are both essential? It's not clear from the presentation of the _HFC experiments whether the Ross suffers the same rapid fate without them.

line 199: I don't think that it would be very clear to a non-specialist reader what is being done here - this is true also of the description of the subsurface warming sensitivity experiment in the next section. Part of the difficulty I think comes from it being rather opaque (yet vitally important to how this modelling system functions) how the ocean-ice coupling actually functions in LOVECLIP without reading the Methods section in detail first. Given my concerns over whether LOVECLIP is actually simulating precipitation over Antarctica well at all, it further not clear that using this to scale the ice melt forcing to the ocean is a very reasonable thing to do either.

line 224: I'm not sure it's true to suggest that this effect is not taken account of in standalone ice sheet model simulations eg Seroussi et al. '21. The CMIP6 climate models used to create the ice sheet SMB forcing in ISMIP6 have snow models as part of their land surface that provide surface melt runoff to the ocean (and thus any cooling that might happen), and some include some sub-surface input of meltwater that parameterises ice shelf melting too. Reductions in surface temperature and surface melt that might be triggered by some ice sheet melting are thus included in at least some of the current CMIP models even without including an explicit ice sheet component.

line 226: it would seem unlikely that a 3 degree ocean model with no ice shelf cavities would be forming Southern hemisphere deep water in a plausible manner however, it's not surprising that no inter-hemispheric interaction between deep water circulations is seen

line 255: the Ross ice shelf collapses by 2100 in your "normal" SSP5-85 simulation too doesn't it?

line 280: the discrimination between scenarios for when accelerated SLR is avoided would seem like a useful finding (although one that can be largely inferred from chapter 9 of the IPCC AR6), and one that I would think this model might be used to investigate with a well-designed search of the parameter space and evaluation against the constraints of historical and modern observations. This also goes for the longer-term, centennial future of the ice sheets that can only be conducted currently with simpler models. Solid earth effects on topographic height and regional sea level become increasingly important, especially for AIS on these timescales.

line 292: comparison should also be made with Muntjewerf et al. 2020 (GrIS) and Siahhaan et al. (2021), who conduct coupled climate-ice sheet experiments to 2100 with the CMIP6 models CESM2 and UKESM1 respectively.

figure 5: what is the subsurface depth range?

figure 6: a horizontal scale on panels a-l would be useful

line 547: what is done with precipitation from the climate model? Is it downscaled, repartitioned?

line 552: an ice sheet horizontal resolution of $\sim 50 \times 100$ km (if I've calculated that right) for GrIS seems very low?

line 557: without specialist knowledge it is impossible to tell what these parameters actually do. Saying that they have been tuned differently from previous studies to compensate for your particular model forcing with no further explanation is not very encouraging

line 570: given how important regional climate details are for ice sheet boundary conditions, I think much more needs to be said about these bias corrections and why one should have confidence that they are equally valid for preindustrial, modern and future climates

line 575: bedrock evolution is mentioned here, I think for the first time. This can be important for AIS grounding line solutions, and more detail would be appreciated. Is the full global sea-level equation solved. Are a range of mantle viscosities taken into account?

line 582: not much is said about what is trying to be achieved with this spinup? It looks like something in equilibrium with preindustrial climate is intended - since the ultimate aim is a transient simulation through the historical and into the future would it not be better to start from a Last Glacial Maximum equilibrium and integrate forward?

line 608: The SSP scenarios are defined from 2015 onward, before that is the historical forcing.

line 625: this paragraph involves a description of how the ocean forcing of the ice shelves is done in the coupling, I think this should be earlier in the Methods

Reviewer #1

The manuscript of Park and others presents results of a climate model of intermediate complexity (EMIC) where the climate model interacts with both the Greenland Ice Sheet and Antarctic Ice Sheet. Compared to former similar studies (Goelzer et al. 2010), they use the Penn State University Ice Sheet/Shelf model (PSUIM), which allows studying the impact of various mechanisms determining, in particular, the ice loss of the Antarctic Ice Sheet, such as the hydrofracturing, Marine Ice-Sheet Instability (MISI), and Marine Ice Cliff Instability (MICI). Since MICI is hotly-debated Marine Ice Cliff Instability (MICI), e.g., (Pollard, Deconto, and Alley 2015; Edwards et al. 2019; Bassis et al. 2021), this study allows determining its impact in an interacting climate context.

To explore the potential sea-level contribution from Greenland and Antarctica, the authors create an ensemble of 10 simulation members for various future climate scenarios ranging from SSP1-1.9, which is a strong climate mitigation scenario, to the high-end scenario SPP5-8.5. The exploitation of the ensemble enables the authors to present the variations due to differences in the initial climate model conditions. More importantly, they perform an extensive set of parameter studies to assess the impact of parametric choices in determining the sensitivity of the ice sheets' sea-level contribution. They analyze the sensitivity of hydrofracturing, MISI, MICI, and an amplified subsurface ocean warming. The results show clearly that the full interaction between ice sheets and the common climate system delays global warming. Furthermore, although the debated MICI influences the results in the coupled simulations, it is not decisive. Ultimately, the authors reveal that only the most substantial climate mitigation scenario SPP1-1.9 avoids a long-term sustained sea-level contribution from Greenland and Antarctica.

This study is highly relevant for several reasons.

1. The missing feedback loop in so-called standalone ice sheet simulations may create misleading projections about the sea-level contribution of Greenland Antarctica, which has numerous implications for the political and social spheres.
2. Various ongoing activities of coupling ice sheet models into state-of-the-art climate models will benefit from the knowledge gained by studies utilizing EMIC because extensive parameter studies, as presented here, are computational not feasible for these models.
3. This study does not settle the debate about MICI. Still, it may lead to a different angle of the discussion because the inclusion/omission has a relatively low influence in the presented simulations compared to other uncertainties.

It was a pleasure to read this excellent manuscript. The study is well organized and written. The figures are generally of high quality, necessary, and informative.

I recommend the publication of the manuscript after a few minor corrections.

Response: We are deeply grateful to the reviewer for reading this manuscript thoroughly and providing insightful and constructive comments which are helpful for improving our work. In this document, we present point-to-point responses to the reviewer's comments.

General comments

The initial conditions of the ice sheet state strongly determine their mid-term (century) evolution. Therefore, I understand that the authors have utilized an asynchronous coupling between the climate model and the ice sheet during the spin-up to cover 8000 years for each ice sheet. Considering the timing of glacial-interglacial cycles (120000 years) or the time since the last glacial maximum (LGM) of about 21000 years before the present, this ice sheet spin-up seems relatively short. Would you please add some more information about the spin-up of the ice sheet into the Method section, such as the initial temperature field and the used starting geometry? In addition, I would like to ask how your results would differ if you would let your model system run an entire glacial-interglacial cycle to spin up the ice sheets?

Response: We agree that initial conditions with longer ice sheet spin-up would be desirable for longer-term projections. In fact, our next step will try to address the initial condition dependency of the ice sheet and sea level projection with much longer spin-up based on more comprehensive coupled ice sheet and Earth System model. Running a full glacial-interglacial cycle with a 3-dimensional fully coupled climate ice-sheet model is a major challenge. To the best of our knowledge, no modelling so far succeeded in this task. Unfortunately, such a daunting task would be beyond the scope of our current paper and it would require large multi-parameter sensitivity simulations. Our current study focuses on understanding processes of climate-ice-sheet dynamics by coupling for the next 100 years. In this target period, our major results would not be critically affected by the spin-up time. As the manuscript figure 2 g and h show, both the GrIS and AIS are stable from 1850 to 1950, indicating that the ice-sheets were at or near equilibrium at the beginning of the PI period. Therefore, we think that the 8,000 years spin-up was sufficient to reach equilibrium conditions. If the 8,000 years spin-up was short to get equilibrium, we would expect to see some drift in the ice-sheets from 1850. For your information, additional figure 1-1 and 1-2 show global surface temperature and ice height of GrIS and AIS at the starting point and ending point (after 8,000 years) of spin-up simulation, and the difference between the starting point and the ending point. Those additional figures have also added as supplementary figure S10.

Additional Figure 1-1 | Global temperature of pre-industrial spin-up simulation. (a) is the starting point and (b) is the ending point (8,000 years) of spin-up simulation. (c) is the difference between (b) and (a).

Additional Figure 1-2 | GrIS and AIS of spin-up simulation. (a) and (b) are the GrIS at the starting point (a) and the ending point (b) of spin-up, (d) and (e) are same as (a) and (b), but for the AIS. (c) and (f) are the differences between them.

For Antarctica and, in particular, Greenland, the surface mass balance is central to the evolution of these ice sheets because it determines where and how much ice is essential gained or lost at the ice sheet-atmosphere interface. Your atmospheric model has a relatively coarse resolution, while the current ablation zone ("melting zone") has a width of several 10 km. Could you please share some information about the surface mass balance pattern that the Greenland ice sheet model receives? How does your downscaling work? You may cite explicitly relevant publications in the Method section to help the reader.

Response: As the reviewer indicated, the atmospheric model included in LOVECLIM has a much coarser horizontal resolution compared to other climate CMIP-type models. Also, because PSUIM considers ice changes not only in the GrIS region, but rather for the whole Northern Hemisphere area, we must rely on rather coarse horizontal resolutions in the ice-sheet model in the northern hemisphere. In the coupling algorithm, surface air temperature and precipitation from LOVECLIM are downscaled vertically to the PSUIM grid with applied lapse rate corrections. We described this at line 70 of the revised manuscript with reference as follows: "Surface air temperature and precipitation are downscaled vertically to the PSUIM grid with applied lapse rate corrections²⁸." The surface mass balance pattern of GrIS is included as an additional figure 2. There are less ice-shelves over the Greenland region, additional figure 2 looks similar to figure. 3a in the manuscript.

Surface mass balance of GrIS

Additional Figure 2 | Surface mass balance of GrIS. Time series of the annual mean GrIS net surface mass balance in SLE. Solid lines indicate the ensemble mean and shading the ensemble range. Different colors represent the historical (black line; period 1850-2014) and SSP1-1.9 (blue line), SSP2-4.5 (pink line), SSP5-8.5 (red line) and SSP5-8.5_MWOF (green line) simulations during the period 2014-2150.

The current partitioning of Antarctica's ice loss ranges from 1:1 to 2:1 for basal melting of ice shelves and iceberg calving (Depoorter et al. 2013; Liu et al. 2015; Rignot et al. 2013). Since iceberg calving contribution seems to be much higher in your simulations than observational estimates suggest, do you have an explanation for the shift towards calving or why the basal melting is too low? Would an artificially amplified basal melting and damped calving be possible to reproduce observational partitioning? If it would be possible, how would it impact the results if you would run your "retune" model system accordingly?

Response: Our model simulation also shows 1:1 ratio for the observation period as indicated in additional figure 3. However, our LOVECLIP simulation suggests that the ratio of ice collapse under global warming would be changed. We also answered your question with regard to sensitivity experiments in the #1-7 special comment.

Additional Figure 3 | The basal melting and ice calving over Antarctica. AIS mass balance terms during (a) 2000 to 2020 and (b) 2080 to 2100 simulated by historical and SSP5-8.5 scenario. Red and blue lines indicate basal melting and ice calving, respectively.

Specific comments

Main document

1-1. Page 2, Line 24: I guess your mean: “Ocean-ice-shelf coupling enhances ...”

Response: Not only ocean-ice-shelf, but also feedback of ocean-ice-sheet is important. Thus, we revise the text at line 23 of the revised manuscript as follows: “Antarctic ocean-ice-sheet-ice-shelf interactions enhance future subsurface basal melting...”

1-2. Page 4, Line 74-88: Here, you talk about the climate fields received by the ice sheets. Would it be possible to state here that you exploit an anomaly coupling approach to deal with the unavoidable biases in the climate state coming from LOVECLIM?

Response: As suggested, we state bias correction at line 68~72 of the revised manuscript as follows: “Biases of surface air temperature, precipitation and SSO are corrected from LOVECLIM to PSUIM²⁶. PSUIM is forced by surface air temperature, precipitation, evaporation, solar radiation and annual mean subsurface ocean temperature. Surface air temperature and precipitation are downscaled vertically to the PSUIM grid with applied lapse rate corrections.”

1-3. Page 5, Line 94-111: The comparison between GRACE and your simulated total mass balance evolution is an excellent validation. However, after reading this paragraph for the first time, I initially had difficulties understanding the difference leading to the Figure 1 and the Supplementary Figure S3. Could you please clarify this part?

Response: The text and figure caption have been revised accordingly.

The text at line 93~104 of the revised manuscript has been revised as follows: “Here we compare the observed *19-year trend of ice mass balance* from the Gravity Recovery and Climate Experiment (GRACE) for the period 2002-2020 to the corresponding values in forced experiments as well as a 5,000-year-long pre-industrial control run (CTR) (Supplementary Fig. S2) conducted with LOVECLIP (Fig. 1). *In Figure 1, each 19-year chunk of mass balance in CTR is cut after high-pass filtering over than 80 years and then, 19-year trends of nature variability are extracted. Those trends are expressed here in terms of sea-level-equivalent (SLE, 1 m SLE = 3.62×10^{14} m³).* Consistent with the GRACE measurements, changes in the mass balance are calculated only from the grounded parts of the ice-sheets for LOVECLIP. Interannual variability of the mass balance recorded by GRACE and simulated by the forced LOVECLIP experiments *during 2002-2020* fall within the range of natural variability exhibited by the CTR (Supplementary Fig. S3).”

The caption of Figure 1 is modified as follows: “19-year trends of observed and simulated mass balance of GrIS and AIS. (a) histogram of *each extracted 19-year trend of Greenland mass balance after 80 year-high-pass filtering* in CTR (gray histogram) with 95% confidence interval range of CTR (black dashed-line), and observed estimates of *19-years trend*~.”

The caption of supplementary figure S3 is modified as follows: “Supplementary figure S3 | Observed and simulated *interannual* mass balance of GrIS and AIS.”

1-4. Page 7, Line 137-140: I understand that "retuning" parameters are a common approach to ensure a reasonable representation of the conditions obtained from observational estimates. How strong is the effect if you would have used the former set of parameters or if you would have used the former standalone setup of PSUIM?
Response: In Deconto, R. M., & Pollard, D. (2016) paper, the model default values of OCFAC (sub-ice-shelf ocean melting), CREVLIQ (hydrofracturing) and VCLIF (ice-cliff failure) are 1, 100 and 3, respectively. In LOVECLIP with those values, the Antarctic ice-shelves increased too much under the pre-industrial condition. Therefore, we changed OCFAC value to 1.5. We also newly described this at line 582 in Method part of the revised manuscript as follows: "The default parameter values are OCFAC = 1.0, CREVLIQ = 100 m per (m/year)⁻² and VCLIF = 3 km/year⁵. Here we use a value of OCFAC = 1.5, which were chosen such that the Antarctic ice-sheet (AIS) has a realistic extent under pre-industrial conditions and for the corresponding LOVECLIM climate forcing.". Additional figure 4 shows the AIS distribution with default values.

Additional Figure 4 | Ice map over Antarctica with default values of OCFAC, CREVLIQ and VCLIF.

1-5. Page 7, Line 178-181: Is it possible to provide separately the MISI, hydrofracturing, and MICI contribution spatially integrated across Antarctica? If so, would you please be so kind as to add such a figure to the Supplement?

Response: Additional Figure 5 shows individual mass balance terms for AIS similar to supplementary figure S5. Because MISI constitutes of a number of instability processes such as buttressing effect, bedrock slope and so on, it is not easy to perform sensitivity experiments on it. So, we just re-plotted the SSP5-8.5, SSP5-8.5_HFOFF and SSP5-8.5_CMOFF (Supplementary Fig. S5) experiments with different naming as MISI (the black-line), No hydrofracturing (the red-line) and No MICI (the blue-line), respectively. Here, MISI includes hydrofracturing and MICI. Please note that the information requested is already provided by the supplementary figure S5.

Additional Figure 5 | Individual mass balance terms for AIS. (a-d) individual AIS mass balance terms for (a) accumulation, (b) surface melting, (c) basal melting and (d) ice calving in terms of SLE. Solid lines indicate the ensemble mean and shading the ensemble range. Different colors represent the simulation with MISI (black), without hydrofracturing (red), and without MICI (blue), respectively.

1-6. Page 10, Line 226-229: Various processes are discussed, conveying the information about meltwater release around Antarctica to the Northern hemisphere. Is the not detected teleconnection a specific feature of your model system? Causes the coarse resolution in the ocean model component the hardly noticed teleconnection, although Figure S6d indicates some connection at the end of the simulations?

Response: Although LOVECLIP has coarse resolution, we expected a change of GrIS due to the AIS meltwater flux. Supplementary figure S6d shows teleconnection between GrIS and AIS for a short period of time, but we could not find any significant change. We think that a high-resolution climate simulation is necessary to explain those interactions. We have discussed it in the discussion section. Considering your comment, we added a new sentence at line 232 of the revised manuscript as follows: “A higher-resolution climate simulation may be required to explain the teleconnection at the end of 21st century shown in Supplementary Fig. S6d.”

1-7. Page 12, Line 240-244: This result again highlights the importance of the coupling allowing the evolution of feedback. As already mentioned in the general section, I would like to ask, why does calving compensate for other fluxes? Is this compensation an artifact?

Response: As we mentioned in response to the general question, the result is not an artifact, but the ratio of ice calving and basal melting would be changed under future global warming and it looks like ice calving compensates the effect of other factors. Additional Figure 6 shows two sensitivity experiments by different ice calving values. The green-line has the same condition as red-line, but with increased ice calving value (1.5) from 2015. In the figure d, the green-line shows a higher ice-calving than the red-line. But in the figure c, basal melting shows like compensated effect, even though both experiments have the same basal melting parameter value. And in consequence, the total floating ice volume (b) and AIS contribution to sea-level (a) by both experiments show similar patterns.

Additional Figure 6 | Time series of sensitivity experiments of ice calving and basal melting. Red-line indicates same experiments as SSP5-8.5 of the manuscript. Green-line has the same condition as red-line, but with increased calving value as 1.5 from 2015.

1-8. Figure 1, caption: Do I understand you correctly: "... (a) histogram of 19-years trends (2002-2020) of Greenland mass balance in CTR ..."?

Response: This is 19-year trend of mass balance in CTR calculated for each 19-year chunk for whole 5,000 years. As also responded to the comment 1-3, the figure caption is revised to deliver clear explanation as follows: **"19-year trends of observed and simulated mass balance of GrIS and AIS.** (a) histogram of each extracted 19-year trend of Greenland mass balance after 80 year-high-pass filtering in CTR (gray histogram) with 95% confidence interval range of CTR (black dashed-line), and observed estimates of 19-year trend for 2002–2020 from the Gravity Recovery and Climate Experiment (GRACE) (blue dashed-line) and simulated by the forced LOVECLIP ensemble (red line)."

1-9. Figure 3, c-e: Do you simulate ice on Island, Elsmere Island, for instance? What is the domain covering the Northern hemisphere?

Response: PSUIM covers the whole Northern hemisphere not just over the Greenland region. Figure 3 partly shows other regions with ice simulation. However, when we calculated mass balance in sea-level equivalent, only the Greenland part was used.

1-10. Figure 3, g-h: If I understand it correctly, the black line represents the grounding line position. Is it correct that this line has disappeared in part of the Ross Ice Shelf in subfigure h? Would it be possible to draw a (dashed) line following the outer ice sheet/shelf edge in the anomaly plots g-h? Since I find it hard to identify a disintegrating ice sheet.

Response: The grounding line on the Ross ice-shelf in subfigure h has not disappeared, but it is moved more closer towards land. Green-lines are added to represent the ice-shelf edge lines in the figure as requested.

1-11. Figure 5, d: You might have missed the subplot label “d”?

Response: You are right and the label “d” is added accordingly.

1-12. Figure 5, caption: You may want to rephrase: "..., (a) the SO subsurface temperature at 400 m depth, (b) ...”?

Response: It is modified accordingly as follows: “(a-c) annual anomalies (relative to the 1850–1900 CE mean) of (a) the SO surface salinity, (b) the 400m SSO temperature and (c) the surface air temperature averaged between 60°S and 90°S. (d) is the SO sea-ice area averaged between 60°S and 90°S.”. Also, other captions of figures are revised accordingly.

1-13. Figure 6, a-l: Does the vertical black line represents the grounding line position? Please mention in the caption what the vertical black line represents. Since I can hardly identify the ice sheet/shelf shape in my version, would you (a) restrict the depth to -2000 m, for example, (b) change the color of the ice or surround the ice shape with a line? If you do not like to change the color of the ice, you may mark the frontal position with a “*”.

Response: It is modified accordingly as follows: “(a-l) Dashed lines indicate grounding lines. (m-p) time series of annual anomalies (relative to the 1850-1900 mean) of (m) SSO temperature, (n) global surface temperature, (o) SL and (p) AIS net mass balance in SLE.”

Method

1-14. Page 29, Line 553-554: Is the resolution of 20 km sufficient to represent the grounding line migration reasonably?

Response: If we want to simulate all of the smaller glaciers' flow realistically, 20 km may be not be sufficient. But even with a more finite resolution or a high-resolution (such as 1km or shorter) model, there still remain a lot of processes that need to be parameterized. We think that 20 km resolution over Antarctica is good enough to represent grounding-line retreat with ice dynamics through the Schoof flux, as discussed in a plethora of studies by David Pollard (Pollard, D., and R. M. Deconto. 2012, Deconto, R. M., & Pollard, D. 2016). Also, we can see distinct grounding-line retreats in figure3 and figure 6.

1-15. Page 30, Line 575-577: Is the meltwater from the ice surface released into the surface ocean? Is the basal melting of ice shelves also released into the surface ocean? Please clarify these points.

Response: Yes, for both of your questions. This is imperfect, but the model does currently not allow for a more realistic coupling. We also newly described this at line 607 in Method part of the revised manuscript as follows: “Both liquid runoff from the surface and basal melting are released into the surface ocean.”

1-16. Page 30, Line 579-580: Does PSUIM consider the bedrock's elastic and viscose response? Does PSUIM consider differences in the response time of the bedrock topography to load changes between East and West Antarctica? Please clarify.

Response: Yes. Isostatic equilibrium as the bedrock response by the elastic lithosphere is applied. But the model only considers vertical bedrock response, not horizontal response. Also, the response time of the bedrock topography is fixed at

3000 years over all grid points (equation number 33 in Pollard, D., and R. M. Deconto, 2012). We newly describe this at line 596 of the revised manuscript as follows: “LOVECLIM’s surface land-ice cover and orography are updated using the simulated ice-sheet and vertical bedrock evolution from PSUIM which is based on elastic lithosphere response with fixed bedrock response time²⁸.”

1-17. Page 30, Line 586: Do you mean:” with chunk lengths of 50 years for the ice sheet model and 5 …”

Response: Yes. To be clearer, we replaced ice sheet with PSUIM as follows: “In particular, the model has been integrated for 120 chunk lengths with chunk lengths of 50 years for PSUIM and 5 years for LOVECLIM”.

1-18. Page 31, Line 588-589: Can you provide some information about the internal ice-sheet temperature at the start of the scenarios simulations?

Response: The ice model output provides basal temperature data. However, this is only one point data where the ice basal sliding is happened, not an exact internal temperature at special ice altitudes. Additional figure 7 is ice basal temperature at the starting point.

Additional Figure 7 | Basal temperature at the starting point.

Supplement Material

1-19. Figure S4, e-f: You may have used some wrong subfigure labels. Please check.

Response: It is modified accordingly.

1-20. Figure S4, f-g: The sudden drop in the parameter study simulations is probably a stock due to the changed conditions. If so, please mention it in the figure caption. If it is not the case, could you please explain it?

Response: Yes. It is due to the changes in parameters associated with hydrofracturing and ice-cliff failure. We added the information in the figure caption as follows: "The sudden drops seen in (f and g) are due to the changes in parameters associated with hydrofracturing and ice-cliff failure."

1-21. Figure S6, d: Do you know what drives the 50% enhanced calving in Greenland for the amplified ocean temperature around Antarctica at the end of the simulation, while the basal melting in Greenland is nearly identical?

Response: In the supplementary figure S6 d, although the ice calving shows a difference between two experiments during late 21st, the ice calving value (0.005 cm in SLE) is much smaller compared to surface melting (~1 cm in SLE), we couldn't find any distinct change over Greenland region which is almost dominated by surface mass balance.

1-22. Figure S7, h: Are the fluctuations in Antarctica's sea-level contribution for SSP2-4.5 driven by waxing and waning ice shelves and subsequent calving events?

Response: Yes, also with a little grounding-line advance.

1-23. Supplement table: You may add to the table caption the information that you have computed ten ensemble members for each climate scenario.

Response: The information about ensemble simulations is added in the table caption as follows: "Ensembles of 10 members with different initial conditions were simulated for historical and SSP experiments, with the initial conditions taken from the last 100 chunks of the spin-up run".

Reviewer #2

Park et al Future sea-level projections with a fully coupled atmosphere-ocean-ice-sheet model

This paper presents a new set of model simulations for future sea level change under three greenhouse gas emissions scenarios, using a method that includes full ice-ocean-atmosphere coupling. The simulations run to 2150, and are able to capture feedbacks between different model components. The experiments show that combined Greenland and Antarctic ice sheet contributions to sea level by 2150 could reach 1.4 m under a high emissions scenario, or as little as 0.2 m under the most stringent mitigation scenario. The results also suggest that the rate of sea level contribution declines through time as a consequence of atmospheric cooling arising from meltwater feedbacks, compared to uncoupled simulations. The authors conclude that more advanced models, such as theirs, are necessary for realistic estimates of future sea-level rise.

The manuscript is well written and the figures are clear. The Supplementary Information file also provides additional useful figures and data tables. Overall I found the paper interesting and the experiments well formulated, and the results will certainly be of relevance to the wider community. There are only two key points that I feel should be addressed in a revised version.

Response: We are deeply grateful to the reviewer for reading this manuscript thoroughly and providing insightful comments which are helpful for improving our work. In this document, we present point-to-point responses to the reviewer's comments.

2-1) Throughout the manuscript the authors assert that their full coupling is new and that their study is the first to account for the atmospheric cooling effect exerted by

ice sheet meltwater (e.g. I26, I213-4, I222, I320). However, whilst elements of their model design are indeed distinct from what has been undertaken previously, it seems odd to me that, despite being cited once or twice in other contexts, the authors don't acknowledge the simulations we presented in Golledge et al., 2019 which specifically DO incorporate the effects of MW-induced atmospheric cooling on ice sheet evolution, both in Greenland and Antarctica. Ours was an asynchronous rather than fully-coupled set up, but nonetheless, our simulations included atmospheric as well as oceanic temperature perturbations arising from meltwater discharge from the ice sheets:

G19 p67. "In the Southern Hemisphere, widespread air surface cooling of more than 2 ° C and local cooling of up to 4 ° C is evident south of approximately 40° S (Extended Data Fig. 1b)."

G19 p71. "by incorporating ice–ocean–atmosphere feedbacks, our simulations predict substantial atmospheric cooling over mid- to high southern latitudes (Extended Data Fig. 1b), which will offset approximately 0.5–3.5 ° C of the atmospheric warming predicted by CMIP5 models (Extended Data Fig. 2c, d) and thus reduce melt."

Because we incorporated the effects of this atmospheric cooling into subsequent runs with the ice sheet model, we captured the consequences of this feedback in a way that eg Bronselaer et al 2018 had talked about but hadn't been able to do (they ran the GFDL atmos-ocean model but had no dynamic ice sheet component). And of course, as the authors are well aware, much of this simply builds on experiments from Menviel et al 2010, Golledge et al 2014 and others. I think also the most recent DeConto and Pollard simulations (DeConto et al 2021) also include atmospheric feedbacks from freshening:

D21 p88. "Including meltwater discharge in CESM expands Southern Ocean sea ice, stratifies the upper ocean, and warms the subsurface (400 m water depth) by 2–

4 ° C around most of the Antarctic margin in the early 22nd century⁴⁶. Conversely, expanded sea ice suppresses surface atmospheric warming by more than 5 ° C, slowing the onset of surface melt and hydrofracturing in the ice-sheet model. The net result of competing sub-surface ocean warming (enhanced sub-shelf melt) and atmospheric cooling (reduced surface melt) produces a substantial negative feedback on the pace of ice-sheet retreat (Fig. 1h). "

So I think it might be worth revisiting some of the wording in the current manuscript to avoid claiming novelty for this particular aspect.

Response: Thank you very much for this comment. We modified our manuscript accordingly and referred the papers you mentioned. In particular, 'which previously had not been considered' was deleted from the original sentence at L25 and 'contrary to what has been hypothesized from uncoupled experiments and studies focusing only on subsurface temperature sensitivities" was deleted from the original sentence at L223, and so on. At line 218 of the revised manuscript, Golledge et al. (2019) and Deconto et al. (2021) are referred.

2-2) The simulations presented use the LOVECLIM intermediate complexity model, which is a low resolution (3x3 deg) simplified atmos-ocean model. It is widely used for both palaeoclimate and future simulations, and does a reasonably job of capturing first order global responses. However, as the authors note (l132), the model has a relatively low temperature sensitivity, which may affect their results. Although they acknowledge this, I think it might be also worth highlighting that the sensitivity isn't globally uniform, and in the ocean it is especially problematic. For example, under certain circumstances LOVECLIM may simulate a global mean SST warming that is similar to CMIP-class models, but the latitudinal gradients are very different. Typically, LOVECLIM vastly underestimates polar amplification, especially in the Arctic, and often even fails to capture mid or low-latitude SST anomalies. The atmospheric response is somewhat better, but is still too cold pretty much everywhere.

It might be useful, therefore, to mention this in the text, and perhaps show a figure that compares the future climate states from LOVECLIM (without the meltwater effect) alongside the comparable CMIP ensemble mean for the same scenario.

Response: This is a valuable comment, and this aspect is discussed in the revised text with additional supplementary figure S8 which is also shown as Additional Figure 1 below. We checked the sensitivity of LOVECLIM compared to CMIP6 on SSP5-8.5 scenario. It is noted that although LOVECLIM has a lower sensitivity than CMIP6 models, it represents Arctic and Antarctic amplification which are in the range of CMIP6 projections. We described this at line 137 of the revised manuscript: “With lower sensitivity, nonetheless, our LOVECLIP shows both Arctic and Antarctic amplification. On the other hand, CMIP6 models do not show the aspect of Antarctic amplification.”

Additional Figure 1 | Arctic and Antarctic sensitivities compared to global surface temperature increase by the end of 21st century. (a) Arctic amplification of LOVECLIP and CMIP6 under the historical and SSP5-8.5 scenarios. Δ temperature is the anomalous mean surface temperature in 2090-2100 relative to 1850-1900. (b) same as (a), but for Antarctic amplification. The Arctic region is defined as latitudes from 60°N to 90°N, and the Antarctic region from 60°S to 90°S.

On the whole though it's a nice paper and an important contribution to the literature.

Nick Golledge

30th May 2022

Reviewer #3

Summary

Park et al have conducted simulations of future climate change using an intermediate complexity Earth System model that includes an ice sheet model component. Using this, they present projections of the contributions of mass loss from the Greenland and Antarctic ice sheets to global mean sea level rise under several different scenarios, with a focus on the period 2015-2150. They also look at some of the impact of omitting aspects of their parameterisation of the coupling between the Southern Ocean and Antarctic ice sheet. Although there are some novel aspects to Park et al.'s approach, their projections are based on a model with little explicit representation of many important physical processes and they show little to convince a skeptical reader that its performance is adequate in a number of crucial areas. There are many aspects of the behaviour of continental ice sheets and their interactions with the climate system that are poorly understood and practically difficult to model realistically; consequently the community's best efforts at making near-term quantitative projections of ice mass loss under future climate change come with extremely large uncertainties (Goelzer et al. 2021, Seroussi et al. 2021). Unfortunately, Park et al.'s simplistic experiment design hardly considers the manifest uncertainty present in our knowledge of the climate-ice system in general and their model in particular. Given the current state of the science in this area, unfortunately I do not think that this study presents projections that can be considered robust or that the analysis they present adds much to our understanding of the relevant parts of the climate system.

Response: We are deeply grateful to the reviewer for reading this manuscript thoroughly and providing insightful comments which are helpful for improving our

work. We understand the reviewer's concerns regarding the use of intermediate complexity models with relatively low resolution. We agree, of course, that the final course of action should be to develop and use highly resolved fully coupled bi-hemispheric ice-sheet-climate models, rather than coarse-resolution coupled models (which may misrepresent high-resolution processes) or forced uncoupled ice-sheet models (which lack important coupled feedbacks and therefore misrepresent sensitivities). Unfortunately, fully coupled high-resolution climate models which resolve all feedbacks and small-scale processes, are not available at this stage. We therefore focus our study on the role of ice-sheet-ocean-atmosphere coupling, and the possibility for compensating feedbacks. Our study provides new insights into the importance of full-component coupling in future climate, ice-sheet and sea level projections.

In this document, we present point-to-point responses to the reviewer's comments addressing those concerns.

General Comments

Understanding and projecting the future ice sheet contribution to sea level rise is a topic of great importance that has seen much progress in recent years via advances in numerical modelling, most prominently perhaps with ISMIP6, the international multi-model community effort to simulate the 21st century evolution of ice sheets consistent with the recent CMIP6 climate projections. If nothing else, the results of ISMIP6 highlight that it is very difficult to robustly model the recent past and near future of the Greenland and Antarctic ice sheets, even with state of the art climate and ice sheet models, and that various groups' attempts to do so lead to a very wide range of possible results. Few of the climate modelling groups that participated in CMIP6 can use explicit coupling to a ice sheet models in their simulations because of the many technical and scientific challenges to doing so plausibly, although two did (CESM2 (Muntejewerf et al. '20) and UKESM1 (Siahaan et al '21) who report on

aspects of next-century simulations coupled to the Greenland and Antarctic ice sheets respectively). In this paper, Park et al. use a model with (when compared to CMIP6 climate models) much lower resolution and low process complexity. Their model includes coupling to an ice sheet component, although their computational shortcuts require the links between climate and the ice sheet model to be heavily parameterised and bias corrected. With this, they have run a small (10 member) initial-condition ensemble of climate change and ice sheet projections and reported on the ensemble average global temperature change and ice sheet mass loss contributions to GMSLR for the same sort of timescale as ISMIP6 did. I'm afraid I simply do not think that decadal-scale projections of the centre-of-probability-distribution global mean sea level rise is a plausible use of this class of model, and Park et al. have not convinced me with this paper.

As alluded to above, the production of climatic boundary conditions for ice sheets is particularly problematic for even state of the art global ESMs. Regional detail in many aspects of both the atmosphere and oceans around the ice is crucial to simulating the evolution of the continental-scale ice sheet. Changes in high latitude atmospheric humidity and circulation, changes in the partitioning of precipitation between snow and rain, the density contrast between open ocean and water on the continental shelves around Antarctica that is itself controlled by the interplay of westerly winds, coastal currents, sea-ice processes and freshwater from the shelves themselves...the list of processes that need to be modelled in an ESM and fed back on as the ice sheets evolve on this timescale is very extensive, and we do not understand many of them well enough to make this sort of projection reliably even in our most sophisticated models. Many of these things seem to be heavily parameterized in LOVECLIP. Such parameterisations can allow unknown processes to be absorbed into a framework where variation and sensitivity can be investigated via different tuning values. However, Park et al. have not done such an investigation. Rather they seem to have tuned their web of parameterisations and feedbacks that contain all this crucial detail to one state and simply run a CMIP6-like set of best-estimate future climate change scenarios. In a field where it is apparent that results

are very heavily model-dependent this single-tuning approach is not likely to produce a robust set of projection numbers, and the lack of process detail contained in their model means that little physical understanding can be gained from analysis of this single configuration of their model. The authors note this heavy qualitative dependence of results on the model and approach to coupling themselves in their Discussion (line 311), but I don't see that they have attempted to do anything to alleviate this weakness in what they have done or the numbers they report.

There is almost nothing to provide confidence that their tuned configuration actually does a reasonable job of simulating the necessary regional detail or sensitivity of almost any aspect of historical climate change or ice sheet evolution. Showing that their ice sheet-integrated mass loss trend to be within the uncertainty range of one short observational record is just not sufficient. Regional features (eg recent ocean warming and increased ice flow rates in the Amundsen Sea region, or the appropriate rate and spatial pattern of surface mass balance at the margin of Greenland) are crucial for determining the near-term evolution of the ice sheets, and there is no evaluation of such factors at all, even though many of these fields should be explicitly present in LOVECLIP rather than wholly parameterised. Some of the results that are shown lead me to think that their model may not in fact be capable of realistically simulating basic important climate processes: it is concerning that the global temperature sensitivity to CO₂ in this tuning is so low, that there is apparently no sensitivity of their Antarctic precipitation to global warming (cf Vignon et al., 2021) and that there are unexplained "bias corrections" used for many of the fields being coupled. None of this means the model is necessarily wrong, but they are all aspects that make it look like this is not an appropriate model to produce a robust multi-decadal projection of overall sea level rise (with only scenario uncertainty reported), or to trust that the component interactions underlying it are happening in a realistic manner.

For projections of the next century, it is important to have both the current state and centennial trends in different aspects of ice sheet evolution correct, not unlike a

weather forecast where the initial conditions of the system matter. Little is said to give confidence that the initial conditions of this coupled system is appropriate, past the evaluation of the twenty year trend for overall ice sheet mass at the end of the century. Appropriate initialisation of coupled climate-ice sheet models is crucial in the projection question, and is an ongoing topic of research. Park et al.'s initialisation strategy is not very clearly laid out, but if it is to aim for reasonably steady ice sheet extents in balance with the preindustrial climate, I would question whether this is an appropriate aim. For the period in question the Earth's ice sheets are still transiently adjusting from the Last Glacial Maximum, so aiming for a balance with preindustrial climate does not seem likely to give rise to an initial state likely to lead to accurate projections, and sampling more members (10 are used) from within this preindustrial simulation doesn't seem likely to cover an appropriate range of possible realistic initial states.

In summary, I think there is not enough shown to have confidence that Park et al. are starting their coupled climate-ice projections from a plausible end of 20C state, nor that the single tuning Park et al have chosen for their model gives rise to an appropriate ice sheet sensitivity to 21C climate change. There is not enough process detail present in their model to learn anything robust about the physical interactions that would happen in reality, and despite some attempt with the briefly-analysed `_MWOFF` and `_HFC` variations, not enough of the model parameter space is explored to give useful quantification of the possible different states of the coupled climate-ice system or strengths of the feedbacks between components that they do have. In my opinion, this is a weak application of an Earth system model of this class. Its strengths are relative computational affordability, which allows for simulations of longer timescales unavailable to more explicit models, or large ensembles that help quantify initial condition uncertainty or the parametric uncertainty in the various component models or the relative importance of different coupling pathways.

I'm afraid do not think this study should be published as it is, and suggest that the authors should radically overhaul their model evaluation strategy and experiment

design if they wish to use LOVECLIP to provide useful information on 21st century sea level rise and climate-ice sheet interactions.

Response: To demonstrate the realism of the Penn-State ice-sheet model in the coupled context, we include an additional supplementary figure, which compares the long-term average ice velocity (1996-2016) observed NSIDC-0484 satellite data with the simulated velocities by LOVECLIP. Our model represents the ice-sheet velocity and shape of ice-shelves over Antarctica in reasonably good agreement with the observations. Points of disagreement include the overestimated size of the ice-shelves and the associated increased velocities in the outflow regions. This additional figure is also added as supplementary figure S11 and we have described this at line 586 of the revised manuscript as follows: “Additionally, LOVECLIP realistically simulates the Antarctic ice velocity and shape of ice-shelves compared to mean of 1996-2016 Antarctic ice velocity obtained from the satellite data (Supplementary Fig. 11). Although, size of the ice-shelves and the associated outflow velocities are overestimated.”

Furthermore, our model captures key instability processes to represent future climate-ice-sheet dynamics, including the previously suggested hydrofracturing and MICI. We agree that the fully coupled climate-ice model should be developed eventually based on a comprehensive Earth System Model and an ice-sheet model with significantly higher resolution. But unfortunately, even going to higher resolution will not necessarily resolve all the parameter uncertainties related to surface mass balance, subsurface melting, ice calving etc. We therefore think, that in the meanwhile, intermediate complexity models can provide new insights into the role of coupled feedbacks involving atmosphere, ocean, ice-sheet/shelf in both hemispheres. They also allow us to systematically study bi-hemispheric ice-sheet instabilities in response to different climate mitigation scenarios and in response to different initial conditions. Given their computation efficiency, EMICs are also suitable tools to perform ensemble sensitivity studies of processes such as hydrofracturing, MISI, MICI and subsurface ocean warming. We clearly understand your comment about “computational affordability”, but we think understanding such

fundamental processes comes prior to your suggestion. And we believe the knowledge gained by the present model will be beneficial to the ongoing activities of developing a fully coupled Earth System Model and ice-sheet model.

Additional Figure 1 | Ice velocity over Antarctica. (a) Annual average of the 1996-2016 ice velocity over Antarctica observed by (a) ICESat and simulated by (b) LOVECLIP.

Specific comments

3-1. line 1: "fully coupled" is an ambiguous and usually inaccurate term for any model of the Earth System, including this one. Many possible component couplings are simplistically parameterised or omitted entirely, and if "fully" is meant simply to indicate that all the components that the modellers wanted to include are permitted to exchange information, it is vague and subjective.

Response: As the reviewer indicated, the use of 'fully' is subjective and it is impossible to develop a fully coupled model in all aspects of the climate system. Therefore, we deleted the "fully" in the revised manuscript, also at the title.

3-2. line 19: Although these interactions are undoubtedly important, I do not think that adding additional complexity to our modelling of elements of the Earth System systems that are as yet poorly understood necessarily leads to more /realistic/ estimates.

Response: We respectfully disagree. Our model results highlight (within the given parameter uncertainties, which also prevail in other higher resolution ice-sheet models) that coupling is a first order process. We argue that omitting coupling in forced ice-sheet models is a major drawback, which renders their projection results unrealistic. Of course, the final solution would be to combine coupling and higher resolved physics, but such models are simply not available at this stage. Until they are, we need to run a two-pronged approach and our higher component complexity modeling study provides a good justification for using coupled (albeit coarser resolution) models in parallel with low-complexity high-resolution modeling approaches. There is no need at this stage to limit ourselves to one path only.

3-3. line 23: "next 130" is inaccurate, given the current date

Response: To clarify, we replace it with "by 2150" at line 22 of the revised manuscript.

3-4. line 23: Given the well-acknowledged uncertainty in many aspects of projecting future sea level, reporting some uncertainty in these numbers and what it represents in the context of the study would in principle be a good idea.

Response: We have included uncertainty values calculated at 95% confidence interval.

3-5. line 26: I'm not sure of the exact context being implied in "had not been considered", but an increase in stratification and thus a cooling influence on the local atmosphere from Southern Ocean freshwater input is not a new result (eg Stouffer et al. 2007) and is implicit in some of the boundary conditions taken from CMIP models for the ISMIP (Nowicki et al 2016.) exercise, since the GCMs include some of the effects of the changing climate on the ice sheets.

Response: Acknowledging previous findings, we modified manuscript accordingly as follows: “Antarctic ocean-ice-sheet-ice-shelf interactions enhance future subsurface basal melting, while freshwater-induced atmospheric cooling reduces surface melting and iceberg calving.”

3-6. line 60: It reduces the buttressing effect of the ice shelf /on/ the grounded ice
Response: It is revised at line 53 of the revised manuscript as follows: “Subsurface Southern Ocean (SSO) warming enhances sub-shelf melting which can lead to a reduction of the buttressing effect of *ice-shelf on grounded ice*.”

3-7. line 66: Given the uncertainties (amongst others) mentioned on line 46- above and gaps in our fundamental knowledge of these processes, I do not think it is true to imply that such models exist, or that the models that do exist can be said to "capture" these interactions. I would also argue that solid earth interactions that may impact the grounding lines of ice shelves, eg Coulon et al 2021 should not be forgotten in this list.

Response: As the reviewer indicated, there is no such model to *fully* capture all the fundamental processes in the Earth System. Among them solid earth interactions, which are not well represented in LOVECLIP (except for the vertical bedrock response due to lithospheric elasticity), could contribute to grounding line retreat and ice-sheet instabilities. We have added the Coulon et al 2021 reference and provide in our revised manuscript a list of unresolved and potentially relevant processes at line 337 of the revised manuscript as follows: “Additionally, glacial-isostatic adjustment⁵¹ other than vertical bedrock response by elastic lithosphere is not fully implemented in LOVECLIP.”

3-8. line 75: Very little is made of any variation between individual members or why each was chosen, and no attempt is made to put numbers to the uncertainty they think is represented by how they have configured the ensemble. In a field where uncertainty in the response and low probability-high impact events are increasingly seen as vital, I find choosing to create an ensemble by sampling only across

preindustrial climate initial condition uncertainty and then not talking about the results bizarre.

Response: We have included the uncertainty ranges as we answered in the 3-4 comment. The ranges are quite small, so it appears that the model results are not sensitive to initial conditions. Given that ice-sheet dynamics are relatively slow, we choose to separate the initial conditions of the ice-sheet by 100 years, to allow for non-negligible differences in the ice-sheet conditions to develop.

3-9. line 81: I don't think enough has been done within the context of this study to claim that the impact of these enormously uncertain processes have been robustly "quantified"

Response: We agree that figuring the parameter's uncertainty out is also crucial to solving realistic problems. However, please note that our main goal is to quantify some key feedback mechanisms of climate-ice-sheet such as hydrofracturing and ice-cliff melting which are still uncertain and not well addressed.

3-10. line 99: what density is being used in this metric?

Response: We used density of glacier ice as 916.7 kg/m^3 .

3-11. line 105: whilst of course it's better than nothing, I don't think that comparison with a single ice-sheet averaged number from a twenty year observational period is anywhere near sufficient to demonstrate that the simulation of the current state of each ice sheet is realistic or that their sensitivity to a changing climate over the next 150 years is likely to be plausible. Nor does it seriously support the implied assertion that the range of variability simulated in CTR is thus representative of non-anthropogenic ice sheet variability in the post-LGM Earth System.

Response: Given that we compared to observations covering 2002-2020, defining "natural variability" as the variability that occurs over a glacial cycle is inappropriate. The point of that comparison is to assess whether the trend during the observational period are inconsistent unperturbed variability in the model, and can therefore be attributed to anthropogenic forcing.

3-12. line 114: it is not to be expected that either the GrIS or AIS mass should be in balance with preindustrial climate, even neglecting ongoing solid-earth adjustments from the Last Glacial Maximum, so some trend is normal (figure S2). But drawing a linear trend line through what appear to be millennial-scale oscillations in each ice sheet seems unfounded, and the two "sudden" drops in AIS mass in the CTR that do not recover are a bit worrying if you want to claim that the system is in some sense "spun-up"

Response: Yes, it is not perfect, but there is also a substantial difference in scale in the response to greenhouse forcing and the variability in the PI run. The changes in GrIS and AIS shown in figure 2a-d is given in meters while that shown in the supplementary figure S2 is in centimeters.

3-13. line 117: I'm seriously concerned that no evaluation whatsoever is done of the performance of the climate model through the 20th century part of the simulation. As is acknowledged, the year 2100 warmings for each of these scenarios are significantly lower than many of the CMIP models, but some confidence in the climate response of relevant parts of the system could be found in the historical period. The level of polar amplification of the global average warming could be verified and would be important, as could shifts in the westerlies over the Southern Ocean. How on earth does the 3 level QG atmosphere cope with climate forcings such as aerosols and ozone that are important for clouds, radiation etc over Antarctica both in this period and in the future under some SSPs?

Response: LOVECLIM has a coarse resolution, simple climate dynamics and a lower sensitivity of global warming than CMIP6 models (additional figure 2-1). It does not have aerosol processes included, nor does it capture ozone dynamics and its effect on the Southern Hemisphere Westerlies. Simply, it is a Model of Intermediate complexity, and not a Coupled General Circulation Model. Still, our model captures feedbacks that CGCM-forced ice-sheet models do not have. Furthermore, many 21st century forced ice-sheet models also use bias corrections in temperature and precipitation, because of large prevailing modeling biases even in CMIP6 models over

polar regions. Clearly, LOVECLIM has a lower sensitivity to global warming, but the additional figure 2-1 shows even higher sensitivity over Antarctica than CMIP6 ensemble mean, and the simulated precipitation response still lies within the CMIP6 range.

Additional Figure 2-1 | Surface air temperature and precipitation over Antarctica.

Annual anomalies (relative to the 1850-1900 mean) of (a) surface temperature and (b) precipitation over Antarctica by CMIP6 models (black line) and LOVECLIP (red line).

Additional Figure 2-2 | Global precipitation patterns. Global precipitation of 1995-2014 mean by (a) observation data CMAP, (b) CMIP6 and (C) LOVECLIP.

3-14. line 124: although warmer temperatures would indeed be expected to increase surface melt on Antarctica, it is also expected to significantly increase the amount of precipitation there (e.g. Vignon et al. 2020) and it is common in many climate models for this increase in accumulation to outweigh the increase in melt for at least some part of the 21st century. I'm concerned that the LOVECLIP projections appear to feature next to no increase in AIS accumulation (eg figure 4) for any of the scenarios and that this part of the mass balance is hardly mentioned. The partitioning of increases in precipitation between snow and rain is crucial too - it is not mentioned how this is handled in the simple 3 level atmosphere or how it changes in the future.

Response: The additional figure 3 show the increasing precipitation under global warming. The decreasing accumulation on the AIS (Fig. 4) is because the Southern surface atmosphere temperature is higher than 0 °C (Fig. 5c).

Additional Figure 3 | Time series of the precipitation anomaly averaged over 70°S-90°S from 1850 to 2150 relative to 1850-1900.

3-15. line 171: it's more than "could be important" - from everything that is known the response of the AIS shelves to warming in the Southern Ocean is acutely sensitive to the regional topography, circulation and salinity as well as temperature. Nothing is shown of whether the modern open ocean state or the coupling method reproduces for instance, the currently observed state of the individual embayments that are expected to be crucial for West Antarctic stability, so it's very difficult to have any confidence in the response of the model. The fact that these processes are entirely dealt with in LOVECLIP via extrapolation of bulk ocean temperature is a

major feature of a model being used for near-term projections in this way and should not be mentioned lightly in passing as it is here. Simply doubling the temperature sensitivity of the coupling parameterisation as in experiment x2_SOTA does not begin to address the uncertainty in this aspect of their model projection.

Response: To address your concern, we emphasized the caveat regarding our model at line ~ as follows: “, which are important to explicitly resolve basal melting processes. In our modeling framework basal melting is parameterized using open ocean temperatures interpolated on the finer ice-sheet model grid,”

3-16. line 174: what processes are involved in this disappearance? This seems an incredible amount of ice to lose on an 80-year timescale, and if it is to be a feature of a model simulation we are meant to rely on for projections it needs to be discussed more. Since LOVECLIP involves somehow-tuned parameterisations of both MISI and the speculative MICI, is it to be assumed that these are both essential? It's not clear from the presentation of the _HFC experiments whether the Ross suffers the same rapid fate without them.

Response: Additional figure 4 is ice thickness of SSP5-8.5, SSP5-8.5_HFCMOFF and SSP5-8.5 minus SSP5-8.5_HFCMOFF in the year 2120. Without hydrofracturing and MICI, there is a bit of delayed ice collapse, but not big difference between them. It means that we will see large ice-sheet melting by future global warming, even without newly suggested hydrofracturing and MICI processes.

Additional Figure 4 | Antarctic ice thickness. Antarctic ice thickness by (a) SSP5-8.5, (b) SSP5-8.5_HFCMOFF and (c) SSP5-8.5 minus SSP5-8.5_HFCMOFF in the year 2100.

3-17. line 199: I don't think that it would be very clear to a non-specialist reader what is being done here - this is true also of the description of the subsurface warming sensitivity experiment in the next section. Part of the difficulty I think comes from it being rather opaque (yet vitally important to how this modelling system functions) how the ocean-ice coupling actually functions in LOVECLIP without reading the Methods section in detail first. Given my concerns over whether LOVECLIP is actually simulating precipitation over Antarctica well at all, it further not clear that using this to scale the ice melt forcing to the ocean is a very reasonable thing to do either.

Response: We modified the sentence about SSP5-8.5_MWOF experiment at line 204 of the revised manuscript as follows: “we performed idealized SSP5-8.5 ensemble sensitivity experiments in which the freshwater coupling from the Antarctic meltwater is decoupled (experiment SSP5-8.5_MWOF).”

Also, we added the equation and sentence about SSO warming experiment at line 254 of the revised manuscript as follows:

“SSO temperatures in the Antarctic ice model T^{IM} are calculated using

$$T^{IM} = 2 \times (T^{LC} - T_{1850}^{LC}) + T_{1850}^{LC}$$

where T^{LC} is the 400 m ocean temperature simulated in LOVECLIM and T_{1850}^{LC} is the corresponding LOVECLIM temperature in year 1850.”

3-18. line 224: I'm not sure it's true to suggest that this effect is not taken account of in standalone ice sheet model simulations eg Seroussi et al. '21. The CMIP6 climate models used to create the ice sheet SMB forcing in ISMIP6 have snow models as part of their land surface that provide surface melt runoff to the ocean (and thus any cooling that might happen), and some include some sub-surface input of meltwater that parameterises ice shelf melting too. Reductions in surface temperature and surface melt that might be triggered by some ice sheet melting are thus included in at least some of the current CMIP models even without including an explicit ice sheet component.

Response: As the reviewer indicated, two-tier approaches on addressing the effect have been done. However, our study tries to address the effect in a coupled manner since positive feedback loops may play out.

3-19. line 226: it would seem unlikely that a 3 degree ocean model with no ice shelf cavities would be forming Southern hemisphere deep water in a plausible manner however, it's not surprising that no inter-hemispheric interaction between deep water circulations is seen

Response: Not sure whether this should be surprising or not, but we still think it is worth reporting. We also agree that it would be interesting to repeat the experiments with a coupled model that can simulate circulation in ice shelf cavities.

3-20. line 255: the Ross ice shelf collapses by 2100 in your "normal" SSP5-85 simulation too doesn't it?

Response: In the figure 6, the Ross ice-shelf collapses only in Re_SSP5-8.5_2xSOTA by 2100 (figure 6h). In the other two experiments (SSP5-8.5 and Re_SSP5-8.5_2xSOTA_MWOFF), the part of ice-shelf still remains by 2100 as seen in figure 6d and l. At line 174 of previous manuscript version, we wrote as "Ross ice-shelf completely disappears in the SSP5-8.5 scenario around 2100" and probably that caused the confusion. We changed the wording "around 2100" to "after 2100" at line 181 in the revised manuscript.

3-21. line 280: the discrimination between scenarios for when accelerated SLR is avoided would seem like a useful finding (although one that can be largely inferred from chapter 9 of the IPCC AR6), and one that I would think this model might be used to investigate with a well-designed search of the parameter space and evaluation against the constraints of historical and modern observations. This also goes for the longer-term, centennial future of the ice sheets that can only be conducted currently with simpler models. Solid earth effects on topographic height and regional sea level become increasingly important, especially for AIS on these timescales.

Response: As the reviewer indicated the discrimination between scenarios for when accelerated SLR is avoided can be inferred from Chapter 9 of the IPCC AR6. However, the assessment in the Chapter 9 is based on mainly two-tier approaches available. We think it is important to show the scenarios discrimination based on the one-tier approach. Regarding the second point, we agree with the reviewer that this would be important subjects for future studies.

3-22. line 292: comparison should also be made with Muntjewerf et al. 2020 (GrIS) and Siahaan et al. (2021), who conduct coupled climate-ice sheet experiments to 2100 with the CMIP6 models CESM2 and UKESM1 respectively.

Response: As following your suggestion, we referred those papers at line 60 of the revised manuscript as follows: “To quantify the effect of these interactions on future SL projections, one needs to employ coupled global climate-ice-sheet models^{24,25}”, because they also emphasize coupling system to project more precisely.

Also, we include and compare to Siahaan’s paper at line 324 of the revised manuscript as follows: “A more dramatic fall in SL, 22 mm by 2100 in the SSP5-8.5 scenario, was found in the U.K. Earth System Model coupled to BISICLES dynamic ice-sheet model simulation²⁵.”

Muntjewerf (2020) studied coupled model, but only the GrIS was coupled. Therefore, we do not compare Muntjewerf’s result in our manuscript.

3-23. figure 5: what is the subsurface depth range?

Response: The subsurface depth is 400 m. This information is added to the figure caption.

3-24. figure 6: a horizontal scale on panels a-l would be useful

Response: We provided the information of horizontal scale as supplementary figure S9.

3-25. line 547: what is done with precipitation from the climate model? Is it downscaled, repartitioned?

Response: The precipitation is downscaled. We also described this at line 68 of the revised manuscript as follows: "Biases of surface air temperature, precipitation and SSO are corrected from LOVECLIM to PSUIM. PSUIM is forced by surface air temperature, precipitation, evaporation, solar radiation and annual mean subsurface ocean temperature. Surface air temperature and precipitation are downscaled vertically to the PSUIM grid with applied lapse rate corrections.", also at line 600 in Methods of the revised manuscript as follows: "Surface air temperature and precipitation are downscaled vertically to the PSUIM grid with applied lapse rate corrections".

3-26. line 552: an ice sheet horizontal resolution of ~50x100 km (if I've calculated that right) for GrIS seems very low?

Response: As the reviewer indicated, the atmospheric model included in LOVECLIM has much coarse resolution compared to other climate models. Also, because PSUIM considers ice changes not only the GrIS region, but the whole Northern Hemisphere area, it has also relatively coarse resolution.

3-27. line 557: without specialist knowledge it is impossible to tell what these parameters actually do. Saying that they have been tuned differently from previous studies to compensate for your particular model forcing with no further explanation is not very encouraging.

Response: We provided the default model parameter values with referred Deconto, R. M., & Pollard, D. (2016) paper at line 582 of the revised manuscript as follows: "The default parameter values are OCFAC = 1.0, CREVLIQ = 100 m per (m/year)⁻² and VCLIF = 3 km/year."

3-28. line 570: given how important regional climate details are for ice sheet boundary conditions, I think much more needs to be said about these bias corrections and why one should have confidence that they are equally valid for preindustrial, modern and future climates

Response: Clearly, we cannot be certain that any model that is tuned for preindustrial or modern day climate (and not entirely derived from first principals) simulates future climate realistically. Unfortunately, we have to rely on this method to simulate a reasonable pre-industrial and present day climate. To the best of our knowledge there is no simple way to justify the use of bias corrections for past or future climate conditions. The same holds all for all empirically determined parameters in climate or ice-sheet models. The same assumption is made for bias-corrected forced ice-sheet offline simulations. Running an ice-sheet model directly with the output from a GCM without bias corrections usually produces unrealistic present-day ice-sheet configurations.

3-29. line 575: bedrock evolution is mentioned here, I think for the first time. This can be important for AIS grounding line solutions, and more detail would be appreciated. Is the full global sea-level equation solved. Are a range of mantle viscosities taken into account?

Response: Yes. Isostatic equilibrium as the bedrock response by the elastic lithosphere is applied and the ice model calculates sea-level with changed bedrock elevation, but only the vertical bedrock response, not the horizontal response. The response time of the bedrock topography is fixed at 3000 years over all grid points. We describe this at line 603 of the revised manuscript as: "LOVECLIM's surface land-ice cover and orography are updated using the simulated ice-sheet and vertical bedrock evolution from PSUIM which is based on elastic lithosphere response with fixed bedrock response time."

3-30. line 582: not much is said about what is trying to be achieved with this spinup? It looks like something in equilibrium with preindustrial climate is intended - since the ultimate aim is a transient simulation through the historical and into the future would it not be better to start from a Last Glacial Maximum equilibrium and integrate forward?

Response: Running a full glacial-interglacial cycle with a 3-dimensional fully coupled climate ice-sheet model is a major challenge. To the best of our knowledge, no

modelling so far succeeded in this task. Unfortunately, such a daunting task would be beyond the scope of our current paper and it would require large multi-parameter sensitivity simulations. Our current study focuses on understanding processes of climate-ice-sheet dynamics by coupling for the next 100 years. In this target period, our major results would not be critically affected by the 8,000 years spin-up time. As figure 2 g and h show, both the GrIS and AIS are stable from 1850 to 1950, indicating that the ice-sheets are at or near equilibrium at the beginning of the PI period. Therefore, we think that 8,000 years spin-up was sufficient to reach equilibrium conditions. If the 8,000 years spin-up was short to get equilibrium, we would expect to see some drift in the ice-sheets from 1850.

3-31. line 608: The SSP scenarios are defined from 2015 onward, before that is the historical forcing.

Response: The text has been modified at line 638 of the revised manuscript as follows: "LOVECLIP follow the historical from the year 1850 to 2014, and SSP1-1.9, SSP2-4.5 and SSP5-8.5 from the year 2015 to 2150"

3-32. line 625: this paragraph involves a description of how the ocean forcing of the ice shelves is done in the coupling, I think this should be earlier in the Methods

Response: we added a new sentence at line 601 of the revised manuscript as follows: "Subsurface ocean temperature is interpolated under ice-shelves on the PSUIM grid.

REVIEWERS' COMMENTS

Reviewer #1 (Remarks to the Author):

The revised manuscript of Park and others presents the results of a climate model of intermediate complexity (EMIC) where an earth system model of intermediate complexity (EMIC) interacts with both the Greenland Ice Sheet and the Antarctic Ice Sheet.

In my opinion, the authors addressed the main issues raised by the reviewers. I agree with the other reviewers that more elaborated studies are needed to understand better the involved processes, interacting feedback loops, and related uncertainties. In particular, coupled simulations with state-of-the-art earth models that implement processes in the atmosphere, land surface, and ocean differently — not necessarily better — are needed. Therefore, this study will probably not act as a reference for sea level projections. However, it will contribute, in concert with other coming studies, to explore the uncertainty range of sea level projections, which are not only driven by uncertainties in one model configuration. Instead, the uncertainty will probably be assessed across a diverse spectrum of models and their configuration variants. Given the widespread biases in climate conditions in the global climate models around Antarctica, the applied anomaly (flux correction) approach is deemed viable.

This study is highly relevant for these reasons, considering the delayed efforts to couple ice sheets into earth system models in the Ice Sheet Model Intercomparison Project (ISMIP) framework.

This study is highly relevant for several reasons.

1. Missing feedback loops in so-called standalone ice sheet simulations may create misleading projections about the sea-level contribution of Greenland Antarctica, even if the used model system may show a too-low sensitivity.
2. Various ongoing activities of coupling ice sheet models into state-of-the-art climate models will benefit from the knowledge gained by studies utilizing an EMIC. Extensive parameter studies, as presented here, are computationally not feasible for state-of-the-art earth system models.
3. The hotly debated Marine Ice Sheet Cliff Instability (MICI) might be discussed differently because the inclusion/omission has, in comparison to other uncertainties, a relatively low influence in the here presented simulations. This result agrees with M. Morlighem's presentation "Investigating the role of Marine Ice Cliff Instability for glacier retreat in the Amundsen Sea Sector over the next century," which he has given at the ISMASS workshop in Island on the 23. August this year.

Since this revised is well organized and written, and the additional figures address raised issues, I recommend the publication of the manuscript.

Reviewer #2 (Remarks to the Author):

The authors appear to have addressed most of the review queries, so I have no further comments to make.

Reviewer #3 (Remarks to the Author):

Thanks to the authors for considering and replying to my review. I'm afraid we're going to have to disagree on this one - the fundamental issues I raised in my comments the first time round have not been addressed and the replies still haven't convinced me, to quote what I said before, that "decadal-scale projections of the centre-of-probability-distribution of global mean sea level rise is a plausible use of this class of model". For the rest, I could simply repeat my General Comments from the first time round.

Don't get me wrong, I'm not at all against the use of intermediate complexity earth system models in general, and have zero objections to the authors' statement:

"We therefore think, that in the meanwhile, intermediate complexity models can provide new insights into the role of coupled feedbacks involving atmosphere, ocean, ice-sheet/ shelf in both hemispheres. They also allow us to systematically study bi-hemispheric ice-sheet instabilities in response to different climate mitigation scenarios and in response to different initial conditions. Given their computation efficiency, EMICs are also suitable tools to perform ensemble sensitivity studies of processes such as hydrofracturing, MISI, MICI and subsurface ocean warming"

I'm just still far from convinced that the experiment design in this paper takes advantage of their computational efficiency to properly address the range of uncertainties that need to be taken into account when making projections of this complex system, or that their model has the appropriate level of detail to give us significant, physically traceable insight into the role of those coupled feedbacks. For me, too many of their results are critically dependent on tuned aspects of the model whose uncertainties are not considered at all.

I appreciate the more specific revisions that have been in response to my comments though, and I can see I'm in the minority opinion of the referee team so I will (more or less) gracefully retire at this point.